



# FarmConners Market Showcases Results: Wind farm flow control considering electricity prices and revenue

Konstanze Kölle[1], Tuhfe Göçmen[2], Irene Eguinoa[3], Leonardo Andrés Alcayaga Román[2],
Maria Aparicio-Sanchez[3], Ju Feng[2], Johan Meyers[4], Vasilis Pettas[5], and Ishaan Sood[4]

[1]SINTEF Energy Research, Trondheim, Norway
[2]DTU Wind Energy, Technical University of Denmark, Roskilde/Lyngby, Denmark
[3]Wind Energy Department, Centro Nacional de Energías Renovables (CENER), Sarriguren, Spain
[4]Mechanical Engineering, KU Leuven, Leuven, Belgium
[5]Stuttgart Wind Energy, University of Stuttgart, Stuttgart, Germany

**Correspondence:** Konstanze Kölle (konstanze.koelle@sintef.no)

**Abstract.** The EU and UK have made ambitious commitments to decarbonise their economies by 2050 under the Net Zero plans. For this, offshore wind will play a major role, significantly contributing to a paradigm shift in the power generation and greater volatility of electricity prices. The operating strategy of wind farms should therefore move from a power maximisation to revenue maximisation design. Wind farm flow control (WFFC) is a key enabler for this shift through mitigation of wake effects in the design and operation phases. The results of the FarmConners market showcases presented here are the first attempt to economically assess WFFC strategies with respect to electricity market prices and revenue.

Here, we present a conceptual simulation study starting from individual turbine control, and extend it to layouts with 10 and 32 turbines operated with WFFC based on the results of five participants. Each participant belonged to a different research group with their respective simulation environments, flow models and WFFC strategies. Via a comparative analysis of relative WFFC benefits estimated per participant, the implications of wind farm size, the applied control strategy and the overall model fidelity are discussed in zero-subsidy scenarios. For all the participants, it is seen that the income gain can differ significantly from the power gain depending on the electricity price under the same inflow, and a favourable control strategy for dominant wind directions can pay off. However, a strong correlation between income and power gain is also observed for the analysed high electricity prices scenarios underlining the need for additional modelling capabilities to carry out a more comprehensive revenue/value optimisation including lower prices and system requirements driven cases.

## 1   Introduction

The ambitious targets 'Net Zero by 2050' within the EU and UK will be driven, in large part, by a significant rise in offshore wind. The UK alone has committed to 40 GW of offshore wind by 2030, a near four fold increase on its current capacity. The rapid rise in offshore wind will result in a shift from electricity prices driven by fossil fuel prices to those driven by the availability of the renewable energy sources. This is expected to increase price volatility, which will challenge wind farm own-





ers/operators to reconsider their design and operation strategy from power maximisation to revenue maximisation, particularly in zero-subsidy schemes.

Tools to support wind farm owner/operators to cope with this transition have been in development for a number of years within the field of wind farm flow control (WFFC). WFFC is the coordinated operation of wind turbines within a wind farm to serve a common goal by taking their aerodynamic interactions into account. This may include a diverse array of objectives from increased energy production and structural load alleviation to environmental and/or societal impact mitigation (Meyers et al., 2022). The most important benefit of the technology, as assessed by experts in a recent survey (van Wingerden et al., 2020), is the increased energy production. It is also the most studied objective with varying results as reported in extensive literature reviews (see e.g., Kheirabadi and Nagamune, 2019; Andersson et al., 2021; Houck, 2022) and seen in the FarmConners benchmark for WFFC code comparison (Göçmen et al., 2022). However, the translation of potential increase in energy production via WFFC to the economic benefits are yet to be quantified. The FarmConners market showcases are designed to address that gap (Kölle et al., 2020; Eguinoa et al., 2021).

## 1.1 FarmConners market showcases

First of its kind, the FarmConners market showcases allow researchers to demonstrate the economic benefits of their control approaches in simulations using realistic environmental and market conditions based on 2020 and 2030 variable electricity price scenarios. Accordingly, the objectives of the showcases can be summarised as:

– Investigate advantages of WFFC for variable electricity prices

– Demonstrate the value of WFFC in existing and upcoming market scenarios

Investigating the advantages when participating in the wholesale electricity markets will provide insight into how WFFC can contribute to the revenue when the power is sold for variable prices. Demonstrating the value in existing and upcoming market scenarios will help to understand the benefit of WFFC in a future with a higher share of wind energy in the energy mix.

The showcases are based on the TotalControl Reference Wind Power Plant (TC-RWP) with 32 DTU-10 MW wind turbines in a staggered layout (TotalControl, 2018), hypothetically located at the west coast of Denmark, 20 km west of the Horns Rev I offshore wind farm. In addition, a simplified subset of the TC-RWP with 10 wind turbines is provided to facilitate participation using computationally intensive codes. However, the showcases data can be used as input to a farm with any number of wind turbines. Independent of the layout, the wind farm receives the simulated electricity price signal for DK1 in NordPool because of the assumed location off the west coast of Denmark. The day-ahead electricity prices for both 2020 and 2030 scenarios are simulated using the balancing tool chain presented in Kanellas et al. (2020), assuming different energy investments over time. The energy scenarios for 2020 and 2030 are qualitative considering relevant factors such as higher prices for fossil fuels in 2030. However, being academic examples, the time series for 2020 does not coincide with the real day-ahead prices for 2020 for DK1 in NordPool. For both of 2020 and 2030 day-ahead electricity prices, weather data by means of wind speed and direction is simulated for the meteorological year of 2012 at the central point of the TC-RWP. The wind turbines and their aerodynamic effects (wakes) are neglected in the simulations, so the wind inflow can be considered as the free stream one.

More information about the data generation for the FarmConners Market Showcases, including detailed descriptions of the
tool chain and assumptions, is available in Kölle et al. (2020); Eguinoa et al. (2021).

## 1.2 Included showcases and participation

Originally, three showcase sets have been specified to reflect different market and operational situations within 2020 and 2030
scenarios; namely 1) high day-ahead electricity prices, 2) low day-ahead electricity prices, and 3) operation by the transmission
system operator, where the last two include structural load alleviation as a performance indicator (Eguinoa et al., 2021).

Five participants have submitted their results for the FarmConners market showcases. These participants are members in
different research groups, and thus, used different simulation environments, different models for the wind farm flow and
different control strategies. Due to the limited number of participating models capable of providing the relevant outputs for the
alleviation of structural loads, the results presented here are mostly limited to the potential benefits of WFFC during the first
showcase set, i.e. high day-ahead electricity prices. For that showcase set, the highest 25% of the electricity prices (both in
2020 and 2030 scenarios) and its respective binned wind inflow data is used (Eguinoa et al., 2021). The resulting distribution
of the prices with respect to the wind speed and direction bins are presented in Figure 1, and further analysed in Sect. 4.1.

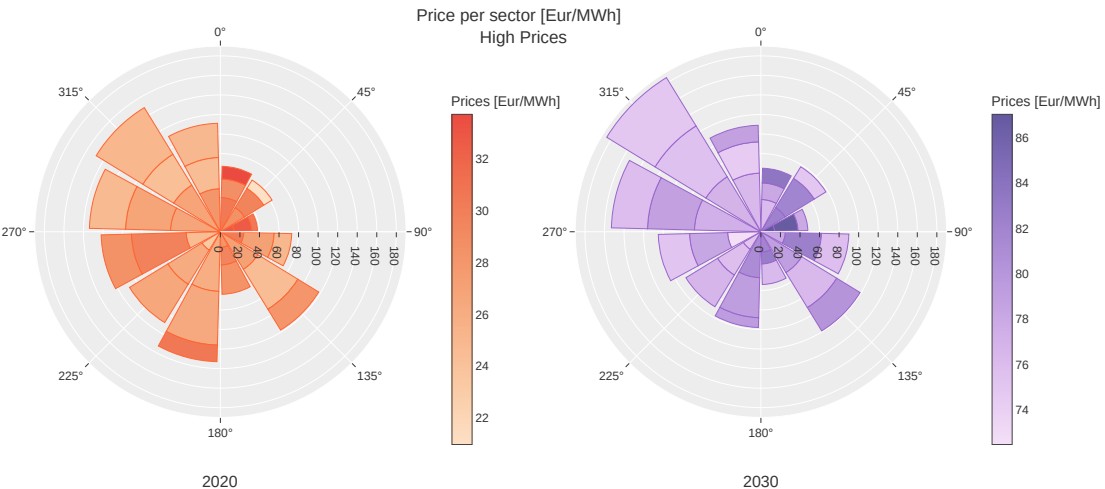

**Figure 1.** Distribution of day-ahead electricity prices per binned wind sector in 2020 (left) and 2030 (right) for the high-prices scenarios
included in the present analysis. Sectors represent the wind direction bins ($\pm15°$). Each sector includes 7 m/s, 9 m/s and 11 m/s ($\pm1$ m/s)
wind speed bins in stacked form following that sequential order from the origin. Radial length of the polar plots indicates the number of
samples per bin. The colour scale represents the prices per bin for both 2020 (left) and 2030 (right) scenarios.

Nevertheless, all three showcase sets are demonstrated for a single turbine with much lower computational cost for aeroelas-
tic analyses, and the extension of the prospects for full-scale WFFC is discussed. Additionally, the differences of the expected
benefits between smaller and larger wind farms are highlighted through the analyses of the subset and full layout of TC-RWP,
i.e. 10 and 32 turbines. The considered setups for all participants are summarised in Table 1.



| Wind farm layout | P1 | P2 | P3 | P4 | P5 |
|---|---|---|---|---|---|
| Single wind turbine | x | | | | |
| Subset of TC-RWP | | x | | | |
| TC-RWP | | | x | x | x |

**Table 1.** Overview of the participants (Participant IDs P1 – P5) per considered layout. The layout of the subset and full TotalControl Reference Wind Power Plant (TC-RWP) with 10 and 32 wind turbines, respectively, is indicated when the results are presented in Sect. 4.

### 1.3 Structure

The structure of the article is as follows. Sect. 2 discusses the methodology and results applied by P1, demonstrating the potential of including revenue based objectives for single turbine control in variable electricity prices of 2020 and 2030 market scenarios. The investigation is then focused on the high price scenario exclusively and extended to farm flow control. Accordingly, Sect. 3 presents the models and methodologies used by the rest of the participants to set up their WFFC simulations for the showcases. A qualitative comparison of the results from these simulations and their corresponding discussion are included in Sect. 4. Finally, conclusions are drawn in Sect. 5.

### 2 Conceptual analysis for a single turbine – Participant P1

The analysis of a single wind turbine, presented in this section, was performed by participant P1. Control objectives of both optimised revenue and reduced structural loads were analysed. It therefore provides deeper insight into the revenue-based control paradigm.

The new method proposed here by P1 is to include the electricity price in the decision-making of the turbine's operational mode. Using down-regulation, power-boosting, and individual blade control (IBC) flexibly according to the instantaneous weather and price conditions, the damage and revenue accumulation of the turbine over time can be managed. Currently, down-regulation (also referred to as curtailment or derating) is used commercially to follow reference power levels according to the grid system requirements. As previous numerical (Pettas et al., 2018) and in-situ studies (Kretschmer et al., 2019) have shown, this process also reduces the structural loading. Moreover, turbine manufacturers already offer turbines with power-boosting capabilities to increase the energy production which comes with a penalty on structural loads. Individual pitch control, in various implementations, has also been extensively researched (e.g., Bossanyi, 2005; Gambier, 2021) and sparsely used in the industry. Combining these existing strategies flexibly and considering the electricity prices and damage accumulation, the revenue and structural load objectives can be managed during the lifetime of the turbine.





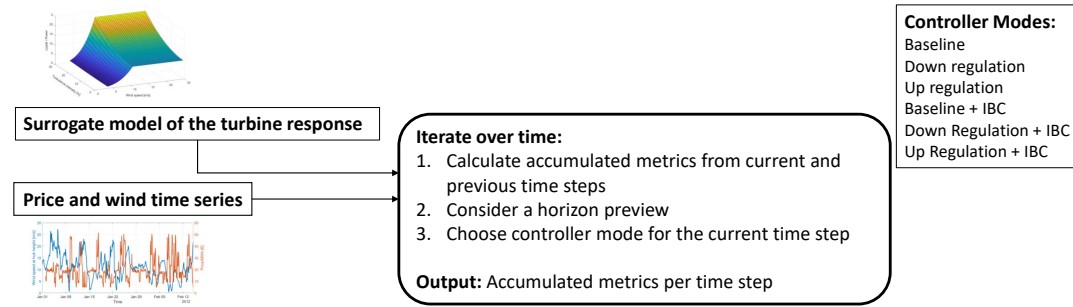

**Figure 2.** Proposed methodology for flexible control operation of a single wind turbine developed and applied by participant P1. IBC: Individual blade pitch control.

The procedure of implementing and evaluating a flexible control method at the wind turbine level is summarised in Figure 2. The baseline turbine controller is modified to include the additional controller modes by adjusting the set-points and tuning the gains. The response of the turbine is derived from aeroelastic simulations and mapped to a surrogate model. This way, the response of the turbine can be quickly evaluated at each time step of the input data (price and weather time series) for a given control mode which can be chosen based on an optimisation logic.

For this study, the baseline proportional integral (PI) controller of the DTU-10 MW reference wind turbine is modified to allow varying nominal power ratings from 5 to 13 MW in down-regulation and power-boost modes. The desired trajectories for these modes are then identified from the maps of power coefficient ($C_p$), tip speed ratio, and blade pitch angle, which are obtained by steady state open-loop simulations with varying wind and rotor speeds. The whole process is automated so that given a desired trajectory, the relevant design variables can be estimated, i.e., the torque constant, rated values, cut in, and rated wind speed. The tuning of the PI pitch controller could not be automated and was fine-tuned manually.

The approach chosen here was to keep the tip speed ratio constant for all power levels and vary the set-points by changing the fixed pitch angle below rated. As other studies have shown (Astrain Juangarcia et al., 2018; D.C. van der Hoek; and Kanev, 2017) there are trade-offs in each chosen trajectory when evaluated concerning load reduction. The IBC scheme is based on the direct feedback of the blade root flapwise bending moment, more details on this design are given in Pettas and Cheng (2018). For the power-boosting mode, the design choice is on whether this should be done with increasing rotor speed or generator torque, or a combination of the two. To avoid inconsistencies at the transition around the rated region, it was decided to keep the tip speed ratio constant at the design level and increase both torque and rotor speed until the new rated value is reached. This contributed to accurate power tracking at the broadest possible range of wind speeds with smooth transition between the modes.

The output of the described procedure is a controller with a required rated value as input and the option to apply IBC. The next step is to create a surrogate model of the response that can be used for evaluation and optimisation. As shown in the literature (see e.g., Dimitrov et al., 2018), different approaches can be followed with trade-offs in computational cost and accuracy. Here, a dense factorial sampling approach was chosen as the dimensions allow for this. The input variable space includes variations of wind speed from 4 to 24 m/s with a step size of 1 m/s and turbulence intensity (TI) from 2 to 24% with a





step size of 2%. The vertical shear coefficient was kept constant at 0.2 to reduce the dimensions as it is less dominant on loads than the other two inputs (Dimitrov et al., 2018). The duration of the simulation was 1 hour and for each case 3 turbulence realisations were considered while yaw misalignment cases were neglected. This set of simulations was repeated for all power 120 ratings between 5 and 13 MW with a step size of 0.5 MW. The IBC cases were evaluated only at wind speeds from 10 m/s and higher. In total, 21420 simulations were performed using the FAST v8 (Jonkman and Buhl, 2005) software to produce the output space.

The mean of the results of the 3 turbulence seeds for each operating point was then tabulated in a 3D matrix for each quantity of interest (e.g., mean of power, DEL of blade root moment, etc.), forming the basis of the surrogate used here. To ensure the 125 smoothness of the model and avoid local fluctuations due to controller tuning or seed-to-seed variability, filtering is applied using a Gaussian convolution kernel. This structure was probed using a spline-based interpolation to produce the surrogate response of the system for any of the quantities of interest.

The evaluation part of the process was implemented similar to a time marching simulation, where at each time step the inputs are wind speed, TI, electricity price, and controller mode and the outputs are instantaneous and cumulative responses. 130 The accumulation of quantities such as energy production was calculated by integrating the values over time. For quantities such as the standard deviation of the rotor speed, the accumulation was done by averaging, and the cumulative load was calculated using equation 1 based on the 1 Hz damage equivalent loads (DEL). $N$ denotes the amount of time steps considered, $y$ is the load channel of interest, $m$ is the Wöhler exponent where a value of 10 is used for the blade related loads and a value of 4 for the rest, and $p$ is the probability of each instantaneous DEL which for this case is equal to $1/N$.

$$\text{DEL}_{\text{cumulative}}(y) = \left[ \sum_{i=1}^{N} [\text{DEL}_{1\text{Hz}}(y_i)]^m p(y_i) \right]^{1/m} \tag{1}$$

The baseline single turbine response was identified by using the entire 1-year time series of the FarmConners market show-cases data set, with the baseline rating of 10 MW without IBC. The optimisation objective is then to reduce the loads and/or increase the revenue over the whole period of each data set (2020 and 2030 time series). This problem is not trivial as there are multiple conflicting objectives in an online optimisation setup. Some of these include the high variation of the sensitivity 140 of the different loads to the controller state and wind speed, the correlation of probability of wind speeds and prices, and the uncertainty of future inputs. For this preliminary application of the method, the problem was simplified and treated as an offline single objective optimisation problem with a perfect preview. The blade root out-of-plane moment (BROop) was chosen as a representative load to be used to correlate power output level to loads for all conditions.

The method was applied for both data sets using the hourly resolution, i.e., the control strategy of the turbine is changing 145 every hour according to the wind and price conditions for the time steps. In Figure 3, an example of operation with different strategies over time is shown for the 2030 data. The controller modes in the y-axis of the upper left plot of Figure 3 correspond to 0 = shut down, 1 = IBC inactive, and 2 = IBC activated.

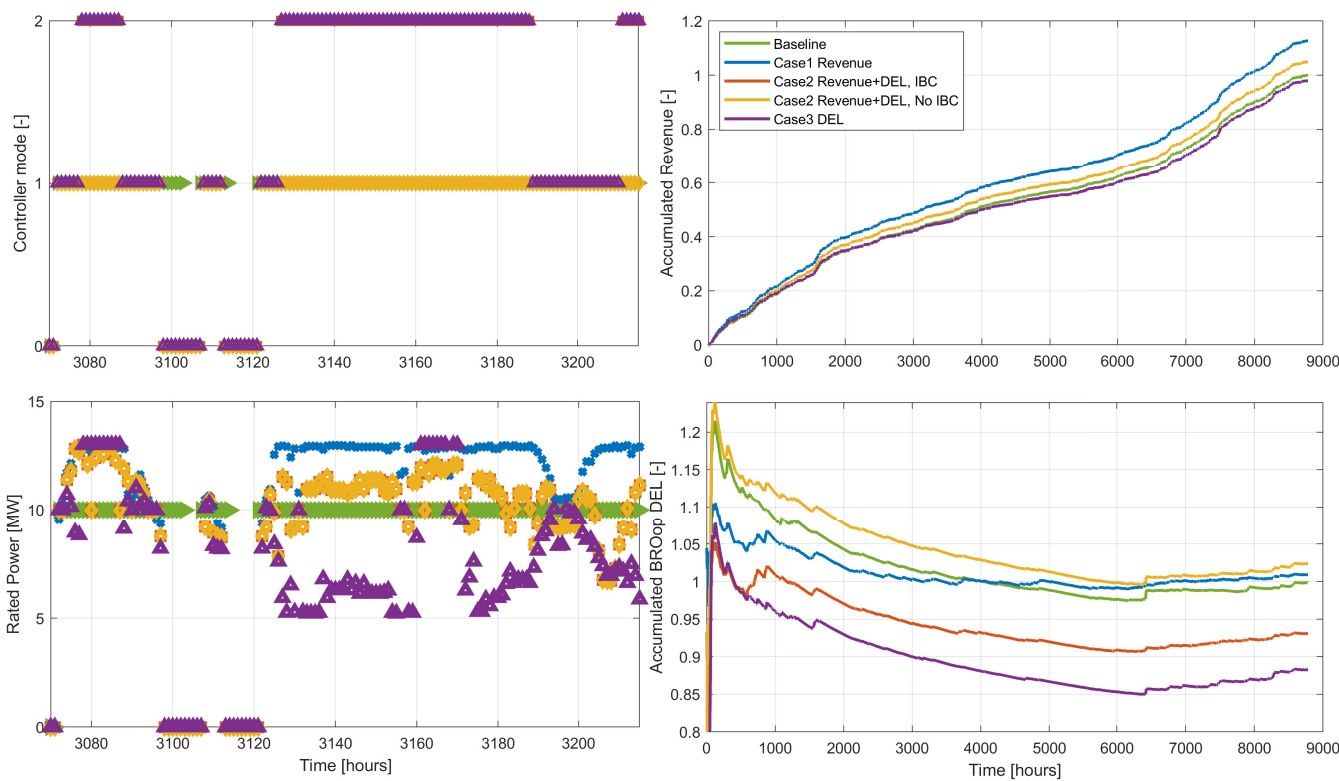

**Figure 3.** Example time series of instantaneous and cumulative values for the one-year time series for 2030. **Left: Upper plot –** Activation of individual blade pitch control (IBC) for controller modes (0 = shut down, 1 = IBC inactive, and 2 = IBC activated) and **– lower plot –** instantaneous power output. **Right: Upper plot –** accumulated revenue over time and **– lower plot –** damage equivalent load (DEL) of blade root out-of-plane moment (BROop) for the same cases, normalised with the final accumulated values of the baseline where the turbine is operated constantly at 10 MW without IBC.

The sensitivity of the method for different objectives is examined with three cases. Case 1 aims to increase the revenue, case 2 to increase the revenue in a load-neutral manner, and case 3 to decrease the loads. In addition, case 2 was run with IBC

and without IBC. In Figure 4 the cumulative results of all methods for both datasets are shown. The trade-off between the two competing objectives is apparent. Case 1 has a higher impact on specific loads which shows that focusing on increasing revenue over some level comes with a high penalty in loads. Comparing case 2 with the IBC switched off and on, we can see the impact that IBC has on specific loads with the most dominant being, by design, the flapwise blade root moment (BRMy). In all cases considering the IBC loop, IBC was activated in power boosting mode and at wind speeds above 13 m/s with TI higher than

3%. The highest negative impact on loads is found at the main shaft torque (LSSTq) and blade root torsion (BRMz), which are increased significantly with power-boosting and cannot be alleviated with down regulation or IBC. The gravity-driven loads of edgewise (BRMx) and in-plane blade root (BRIp) moments are not affected in any way. In both years, the achieved increase in





revenue for case 2 was about 5% with neutral or decreased loads. The maximum revenue was 10% for 2020 and 13% for 2030 with a penalty of 15-20% at the loads influenced most by boosting.

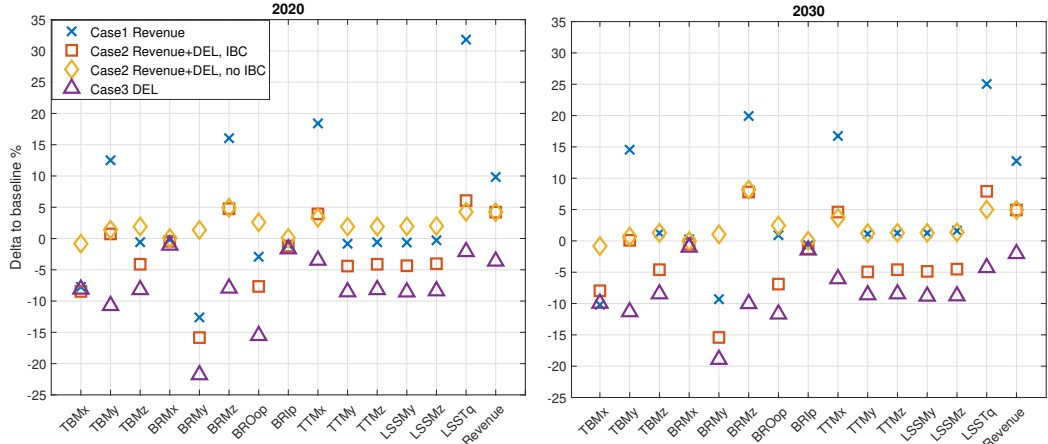

**Figure 4.** Cumulative values of damage equivalent loads (DELs) and revenue for operation of a single wind turbine with different objectives, in relation to the baseline operation. TBMx: tower bottom side to side moment, TBMy: tower bottom fore-aft moment, TBMz: tower bottom torsion, BRMx: blade root edgewise moment, BRMy: blade root flapwise moment, BRMz: blade root torsion, BROop: blade root out-of-plane moment, BRIp: blade root in-plane moment, TTMx: tower top roll moment, TTMy: tower top pitch moment, TTMz: tower top yaw moment, LSSMy: non-rotating low speed shaft bending moment about the Y axis LSSMy: non-rotating low speed shaft bending moment about the Z axis, LSSTq: low speed shaft torque.

The presented results are based on a basic optimisation logic including user-defined thresholds, and trial and error methods. Nevertheless, the results show that there is sensitivity of structural loads and revenue to the method used here, and in other tested cases, not shown here, different trade-off levels could be achieved. Moreover, a different behaviour was observed between the two years. Finding an effective tuning for the 2020 dataset was much harder to achieve than for the 2030 dataset. More threshold values had to be tested and fine adjustments could result in larger changes. On the other hand for 2030, even with

rough estimations, the optimisation objectives were achieved without much tuning.

Since the weather time series are the same for both years, the difference lies in the price patterns. Figure 5 shows the contribution of each binned price and wind speed to the cumulative load and revenue for the two years considering baseline operation. In 2030 the higher variability of prices produces more discrete and uncoupled regions of influence for each objective. This allows to define thresholds optimising the two competing objectives with fewer trade-offs and less detailed knowledge of

the probabilities. This example shows that the boundary condition of the markets in which the wind farms operate, influences significantly how much flexible control (or any other turbine or farm control strategy) can influence the structural loading and revenue of the system. The main impact does not come from the mean level of the prices but their variability over time. For the near-future energy systems with increasing renewable penetration, this behaviour can be expected and the approach shown here can be effective to optimise the operational strategy.





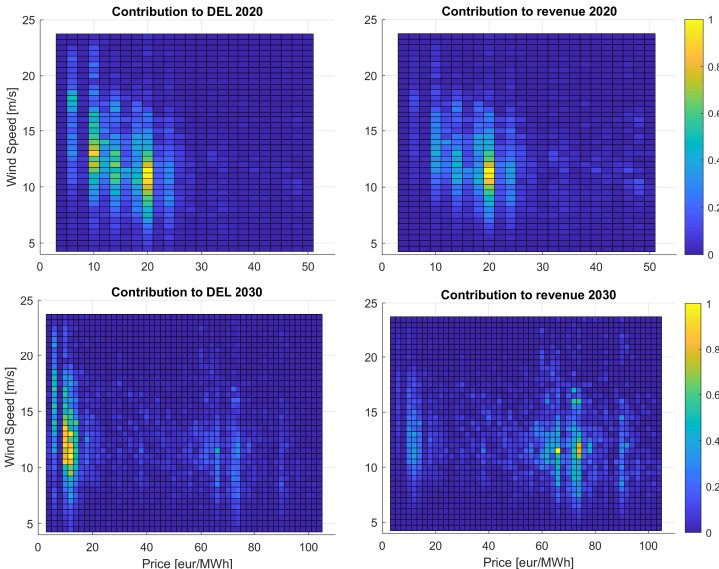

**Figure 5.** Contribution of price and wind speed bins to revenue and damage equivalent load (DEL) of blade root out-of-plane moment (BROop), obtained by multiplying the counts of each bin with the response obtained with the baseline wind turbine controller in simulations with the full datasets for 2020 and 2030. All values are normalised to the maximum value for the specific year and metric. Note that regions with high contributions to revenue on the right are strongly correlated to the bins with (below rated) wind speed and direction in Figure 1 included in the showcase with high day-ahead electricity prices.

The results here are based on the initial application of the proposed method which will be discussed in detail in a separate publication. The next step would be to add the dimensions of wind direction and wind farm (flow) control to evaluate the holistic potential of flexible control in wind energy in the future electricity markets. On that regard, the rest of the paper focuses on the most beneficial price and wind speed bins for wind farm flow control as identified in Figure 5 with corresponding wind directions presented in Figure 1.

**3    Participants methodology for WFFC – Participants P2-P5**

The analysis of the single turbine control for increased revenue by P1 is extended in this section to wind farm level to explore the potential power and income gains of different WFFC oriented models with varying control strategies, both for the 10-turbine layout (participant P2) and the full TC-RWP with 32 turbines (participants P3 – P5). Table 2 provides a summary of the flow models used by the participants. More detailed descriptions of the models and the simulation setups are provided below

for each participant.





| | P2 | P3 | P4 | P5 |
|---|---|---|---|---|
| Wind farm model | FLORIS[a] | PyWake[b] | FLORIS[a] | – |
| Control strategy | Wake steering | Combined[*] | Wake steering | Combined[*] |
| Velocity deficit model | Gauss-legacy[c] | Fuga (linearised CFD) | Gauss-legacy[c] | Gauss-legacy[c] |
| Added turbulence model | Crespo-Hernández[d] | – | Crespo-Hernández[d] | Crespo-Hernández[d] |
| Wake superposition | SOSFS[e] | Linear | SOSFS[e] | Recursive[f] |

**Table 2.** Overview of participating models. [a](NREL, 2021), [b](Pedersen et al., 2019), [c](Bastankhah and Porté-Agel, 2014), [d](Crespo and Hernández, 1996), [e]SOSFS – Sum of Squares Freestream Superposition, [f](Lanzilao and Meyers, 2021). [*]Combined control strategy corresponds to the combination of static wake steering and static axial induction.

### 3.1 Participant P2

#### 3.1.1 Wind farm model

The wind farm model used by participant P2 is FLORIS (NREL, 2021), where the following sub-models have been applied:

- Velocity deficit: Legacy Gauss (Gaussian model), by Bastankhah and Porté-Agel (2014).
- Wake added turbulence: Crespo-Hernández model (Crespo and Hernández, 1996).
- Wake superposition: modelled with SOSFS – Sum of Squares Freestream Superposition, to combine the wake velocity deficits to the base flow field (Katic et al., 1986).
- Wake steering: Wake deflection model by Bastankhah and Porté-Agel (2016).

Additionally, power loss due to yaw misalignment in the controlled upstream turbine is modelled by scaling the effective
wind speed as suggested by Bossanyi (2019) in equation 2, where $v_{avg}$ is the rotor-averaged wind speed, $\Psi$ is the steering control setting and $n$ is the yaw loss exponent. Related power is then obtained based on this scaled wind speed and the power curve.

$$v'_{avg} = v_{avg}cos(\Psi)^{n/3} \tag{2}$$

The values for the corresponding model parameters utilised in the sub-models listed above can be found in Table 3. The
wake velocity and turbulence model parameters have been obtained from previous calibration studies on other offshore wind farms. For $TI_{initial}$, the value of ambient TI has been used. The yaw loss exponent value $n$ is selected within the range of values already reported in the literature —see examples in Simley et al. (2021)—. Shear is considered equal to zero.





|     | $\alpha$ | $\beta$ | $k_a$ | $k_b$ | $TI_{\mathrm{constant}}$ | $TI_{ai}$ | $TI_{\mathrm{downstream}}$ | $n$ |
|-----|------|-------|--------|----------|--------|----------|---------|------|
| P2  | 3.0  | 3.0   | 0.3503 | 0.005312 | 0.1251 | 9.932e−5 | -0.4689 | 1.61 |
| P4  | 0.58 | 0.077 | 0.38   | 0.004    | 0.5    | 0.8      | -0.32   | 1.88 |

**Table 3.** Model parameters for FLORIS used by P2 and P4. P2 used a set of in-house calibrated parameters, P4 the default values. For a description of the parameters, consult the documentation of FLORIS (NREL, 2021).

### 3.1.2  Wind farm and wind turbine control strategies

The wind farm optimisation has been executed using the algorithms available in the FLORIS library, using Sequential Least
Squares Programming (SLSQP). The control strategy at the wind farm level was aimed at maximising the total energy production of the wind farm per bin through yaw steering. The wind rose has been divided into 144 sectors (2.5º steps) and 2 m/s intervals for wind speed. Optimal set-points were obtained individually for each combination of wind speed and direction.

Yaw misalignment for wake steering is limited within the range [0º, 20º], where the sign convention is positive yaw misalignment for counter-clockwise rotation viewed from above. The range is in accordance with results in the literature (Simley
et al., 2021).

### 3.2  Participant P3

#### 3.2.1  Wind farm model

Participant P3 used PyWake (Pedersen et al., 2019) to model the TC-RWP. PyWake can model a wind farm in two ways, considering only downstream wake propagation as well as an iterative method that is able to consider wind turbine blockage
effects. Either approach counts with several wake, blockage and deflection models in its library. The configuration used by P3 is the following (see Table 1):

- Velocity deficit model: Fuga (Ott and Nielsen, 2014). Linearised CFD model developed by DTU. It models wake deficit and wake propagation assuming uniform and fixed turbulence intensity (TI) over the wind farm. In this case a fixed, conservative TI = 5% was assumed.

- Wake superposition: Linear superposition, which is one of the basic assumptions in Fuga.

- Added turbulence: Fuga assumes uniform turbulence represented by vertical shear and surface roughness height. No added turbulence is needed for estimation of wake expansion and deficit decrease over distance.

- Wake yaw deficit model: Fuga Yaw Deficit and Fuga Deflection models. Fuga Yaw Deficit models implicitly the deficit and deflection models, and calculates a sets of look-up tables with the (yaw) longitudinal deficits $UT$ and $UL$, due to
transverse and longitudinal unit forces, respectively. Fuga Deflection, like the Fuga Yaw Deficit, estimates tables for $VT$ and $VL$, the transverse deficits generated by unit forces in the transverse and longitudinal directions. Since Fuga is a linear model, the final deflected deficit is calculated by linear superposition of $UT$, $UL$, $VT$ and $VL$.





### 3.2.2 Wind farm and wind turbine control strategies

The control strategy was applied to wind turbine and wind farm levels.

The control strategy at the wind farm level aimed to maximise the total wind farm power production by individual control of wind turbine´s derating and wake steering. The set-point of each turbine was determined using TOPFARM, the Python package developed by DTU for wind farm optimisation. The routine built a look-up table of set-points (derating and yaw angle per turbine) dividing the wind rose in 120 sectors of $3°$ and incremental steps of 1 m/s. Finally, the set-points for the wind direction and speed in the high prices scenarios for 2020 and 2030 were obtained by interpolating this table.

Wind turbine loads were not estimated by P3, but each individual turbine curve was constructed to operate at maximum power coefficient, $C_p$, at a given wind speed and yaw misalignment, while minimising the thrust coefficient, $C_t$, for loads alleviation. The power curves were constructed using HAWC2aero, subset of HAWC2, and also developed by DTU, to simulate the dynamics of a rigid turbine rotor under aerodynamic loads. This approach allows a more precise estimation of $C_p$ and $C_t$ under yawing conditions.

## 240 3.3 Participant P4

### 3.3.1 Wind farm model

Participant P4 used FLORIS (NREL, 2021) to model the wind farm flow and power production. The default setting of FLORIS was used, which means the following sub-models were activated:

– Velocity deficit model: Legacy Gauss (Gaussian model) by Bastankhah and Porté-Agel (2014).

– Wake added turbulence model: Crespo-Hernández model (Crespo and Hernández, 1996).

– Wake superposition: modelled with SOSFS – Sum of Squares Freestream Superposition, to combine the wake velocity deficits to the base flow field (Katic et al., 1986).

– Wake steering: Wake deflection model by Bastankhah and Porté-Agel (2016).

Default parameters of the FLORIS sub-models were used in P4 simulation, which are also listed in Table 3. Note that for
simulating yawed and non-yawed wind farm production for each inflow bin, the wind shear exponent was derived with a power law based on the mean wind speed at two heights: 50 m and 150 m, while the mean turbulence intensity measured at hub height was used.

### 3.3.2 Wind farm and wind turbine control strategies

Wake steering was chosen by P4 as the control strategy. Considering the range of wind directions per inflow bin and that
different wind directions have different potential for wake steering, P4 took an averaging approach when simulating the wind farm flow and power production, both for the non-yawed cased (Normal operation) and the yawed case (with WFFC).



For a given inflow bin, 31 flow cases were considered, each representing a flow case with a wind direction in the range of [-15º, 15º] around the wind direction bin centre, $\theta_{cen}$, and a wind speed as the mean wind speed, $u_{mean}$, at hub height.

For each of the 31 flow cases, the optimal yaw angle of all turbines was found using the SLSQP (Sequential Least Squares Programming) algorithm included in FLORIS, with the maximal iterations set to 200. The objective of this optimisation problem is to maximise the total power output of the wind farm:

$$\gamma_1^*, \gamma_2^*, ..., \gamma_{N_t}^* = \underset{\gamma_1, \gamma_2, ..., \gamma_{N_t}}{\arg\max}\ P_{tot}(\gamma_1, \gamma_2, ..., \gamma_{N_t} | u_{mean}, \theta) \tag{3}$$

where $\gamma_i^*$ represents the optimal yaw angle of the $i$th turbine, $\theta \in [\theta_{cen} - 15, \theta_{cen} - 14, ..., \theta_{cen}, ..., \theta_{cen} + 14, \theta_{cen} + 15]$ is the wind direction for this flow case, and the lower and upper bounds of yaw angle for all turbines are set to -25º and 25º.

Thus, for each inflow bin, there are 31 sets of optimal yaw angles, each with a different total power output $P_{tot}$. The final reported power output $P_{\text{WFFC}}$ for this inflow bin was the mean of these 31 cases. Similarly, when calculating the normal power output $P_{\text{Normal}}$, the mean of the 31 flow cases was also used.

By considering 31 flow cases with different wind directions and solving the yaw optimisation problem separately for each inflow bin, P4 took an idealised or 'greedy' approach that tends to explore the full potential of wake steering, since the effectiveness of wake steering can be quite sensitive to the wind direction. However, in real-life implementation, limits on the speed and accuracy of the yawing system, uncertainty of the measured inflow wind direction, and other factors can make the reported energy gain hard to be fully realised.

### 3.4  Participant P5

#### 3.4.1  Wind farm model

Participant P5 used an in-house analytical wake model which combines a Gaussian wake model with a recursive wake merging methodology (Lanzilao and Meyers, 2021). The model has previously been used in a power optimisation study for the TotalControl wind farm using wake steering (Sood and Meyers, 2022). The following model specifications were used by P5

- Velocity deficit model: Legacy Gauss (Gaussian model), by Bastankhah and Porté-Agel (2014).
- Wake added turbulence model: Crespo-Hernández model (Crespo and Hernández, 1996).
- Wake superposition: Modelled using a recursive wake merging methodology (Lanzilao and Meyers, 2021).
- Wake steering: Wake deflection model of Bastankhah and Porté-Agel (2016).

#### 3.4.2  Wind farm and wind turbine control strategies

The control strategy applied by P5 was a combination of wake steering and axial induction control of individual wind turbines within the farm to achieve the optimisation objective. For power maximisation, this included finding the optimal yaw and thrust set-points of all the turbines within the farm to achieve maximum power production according to the equation





$$
\begin{aligned}
&\min_{\boldsymbol{\gamma}, \boldsymbol{C_T}} && -1 * \sum_{k=1}^{N_t} \frac{1}{2} \rho C_P(\gamma_k, C_{Tk}) A_k U_k^3(\boldsymbol{\gamma}, \boldsymbol{C_T}), \\
&\text{s.t.} && -\frac{\pi}{6} < \gamma < \frac{\pi}{6}.
\end{aligned}
\tag{4}
$$

where, $C_P$ is the coefficient of power of each turbine, evaluated for a yaw angle $\gamma_k$ according to the cosine power law (Doeke-meijer et al., 2020). The effect of induction control by using a sub-optimal thrust coefficient $C_T$ was determined by using a look-up table which was previously developed using OpenFast. $\boldsymbol{\gamma}$ and $\boldsymbol{C_T}$ are vectors containing the yaw and thrust set-points for all the turbines across the farm. Turbine yaw angles are limited between $\pm 30°$ to prevent excessive fatigue and the thrust coefficient is limited between 0 and 1 to avoid turbine shutdown or over-induction. $A_k$ is the area of each turbine, and $U_k$ is the inflow velocity of each turbine as a function of upstream yaw angles and thrust set-points, determined using the wake model. The optimisation problem is solved to obtain set-points for all the turbines within the wind farm using the SLSQP solver from the SciPy Python package, while utilizing the multi-start approach of basin-hopping to avoid local minima (Virtanen et al., 2020).

## 4  Results & Discussion

The FarmConners market showcases define only the layout of the wind farm and as simulation input the wind speed and direction as well as electricity prices. Any other detail of the implementation is left to the participants. Possible reasons for different results include (1) different models for wind farm flow and power production or different parameters for the same model; (2) different control strategies; (3) different optimisation problem formulation and bounds/constraints on design variables; (4) different optimisation methods or different settings of the same algorithm.

Therefore, the WFFC algorithms detailed in Sect. 3 are not directly compared against each other, but to a wind farm operation without WFFC (also referred to as normal operation) for the same model implementation. During the high price scenarios for both 2020 and 2030, it is expected that maximum power production will be favoured in the interest of generating the highest possible income. Accordingly, the relative metrics for power and income gain are presented in this section for analysis and discussion.

### 4.1  Analysis of weather and price conditions

Before presenting the specific results per participant, a comparative analysis between 2020 and 2030 is performed for the weather and price conditions to support the subsequent discussions. The price time series for 2020 and 2030 were simulated for the same meteorological year assuming different energy scenarios at the systems level (Eguinoa et al., 2021). Moreover, the investigated high-prices showcase includes only wind speeds between 6 and 12 m/s, where the strongest wake effects are typically observed and the highest benefit of WFFC can be expected. Thus, the included wind-speed bins are between cut-in and just above the rated wind speed for the DTU 10 MW reference wind turbine.





As shown in Figure 1, absolute price values are 2-3 times higher in 2030 than in 2020. If wind sectors are analysed, it can be

seen that the occurrences in the North-West quadrant in 2030 increase with respect to 2020, at the expense of the South-West quadrant. Similarly, the highest prices within each year are distributed differently in 2020 and 2030. In 2020, the bins with the highest prices are [15°-centred sector, 11 m/s] and [75°-centred sector, 7 and 9 m/s], followed by [195°-centred sector, 11 m/s]. By contrast, in 2030 the bins with the highest prices are mostly concentrated at [75°-centred sector, 7 and 9 m/s], followed by [15°-centred sector, 11 m/s].

In conclusion, the prices distribution among wind direction and wind speed bins is not equal in both years despite being based on the same meteorological time series. This reflects how the specific combination of site wind conditions and market prices shall impact planned operation and profitability assessments for a particular wind farm as the market evolves during its lifetime.

## 4.2 Power gain

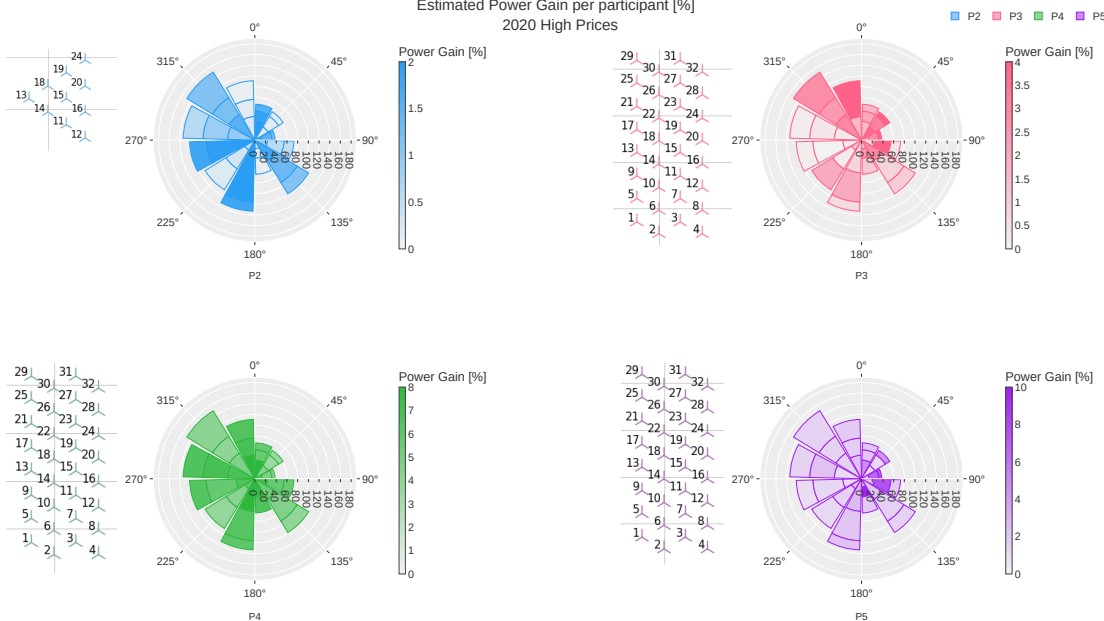

**Figure 6.** Power gain when using wind farm flow control (WFFC) in simulations with the wind inflow and price information for 2020 as shown in Figure 1. Each polar plot presents the results of one participant. The wind farm layout considered by this participant is shown on the left side of the polar plot.

The estimated increased production per bin is normalised with respect to the production under normal operation without WFFC. For each sector, the stacked gains are indicated as heat maps and reported per 7 m/s, 9 m/s and 11 m/s along the radial direction, with respect to the number of samples per bin.

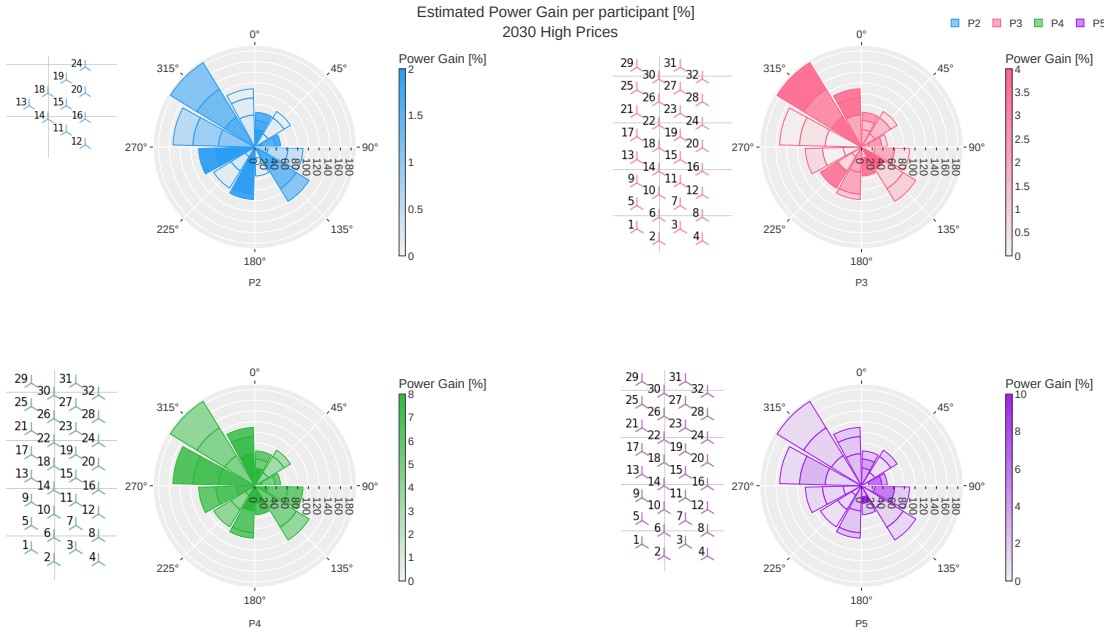

**Figure 7.** Power gain when using wind farm flow control (WFFC) in simulations with the wind inflow and price information for 2030 as shown in Figure 1. Each polar plot presents the results of one participant. The wind farm layout considered by this participant is shown on the left side of the polar plot.

The estimated increased production per bin is normalised with respect to the production under normal operation without WFFC. For each sector, the stacked gains are indicated as heat maps and reported per 7 m/s, 9 m/s and 11 m/s along the radial direction, with respect to the number of samples per bin.

Figures 6 and 7 show the normalised power gains per wind direction for 2020 and 2030, respectively. The results are presented in separate polar plots for each participant with the considered farm layout on the left. Each sector includes the wind speed bins in stacked form in the order 7 m/s, 9 m/s and 11 m/s ($\pm 1$ m/s) from the origin. The radial extension of a bin represents its number of samples, which is fixed in the showcases set for all participants. The heat map indicates the normalised power gain per bin in percentages, as shown in equation 5.

$$\text{Power Gain}_{\text{bin}} = \left[ \frac{P_{\text{WFFC}} - P_{\text{Normal Operation}}}{P_{\text{Normal Operation}}} \right]_{\text{bin}} \cdot 100 \tag{5}$$

where $P_{\text{WFFC}}$ is the farm-wide power using WFFC and $P_{\text{Normal Operation}}$ the farm-wide power in normal operation, calculated as the sum of the power of all wind turbines. Thus, the normalisation accounts for the size of the wind farm but not the particular layout. The total power $P_{\text{WFFC/Normal Operation}_{\text{bin}}}$ generated by the wind farm per bin is the sum of the power production of all samples in that bin.

The results from P2 for 2020 are shown in the upper left polar plot in Figure 6. The heat map represents the estimated power gain where darker shades correspond to higher gains. Accordingly, it shows that power gains of up to 2% are estimated by P2



for the considered 10-turbine layout. Sectors with high power gains alternate with others with lower gains (e.g., 15° shows high power gains being adjacent to the sectors 345° and 45° with low power gains). These high-gain sectors include the wind directions that are aligned with the longest rows of turbines, with higher expected wake losses. Participant P3 reports power gains of up to 4%. The sectors with northwest and easterly wind show higher power gains than the rest. In bins with western wind, very little to no power gain is reported. This division into favourable wind directions with several adjacent sectors having very high or low power gains is most extreme among all participants, indicating high sensitivity to wind direction in the implemented control settings. The green polar plot shows the results from P4 with power gains of up to 8%. The power gains achieved by P4 reach similar values in all sectors having the most uniform distribution across all bins from all participants. Participant P5 reported the highest power gains, per bin, of up to 9%. A few bins with easterly wind have very high power gains although their adjacent bins have very low or no power gain. Adjacent wind speed bins with high and low power gains in the same sector are observed exclusively in P5 results.

The discussed Figures 6 and 7 show consistent power gains per participant for both years. As discussed earlier, the inflow conditions for the simulated energy scenarios are identical. However, the differences are observed due to the definition of the high prices showcases as they correspond to slightly different periods during 2020 and 2030, resulting in nonidentical wind distributions shown in Figure 1. The power gains from P2 for 2030 are very similar to those in 2020. The distribution of power gains between the wind speeds per sector is almost the same. An exception is the 75°-sector, where the 11 m/s-bin has the lowest power gain in 2020 but the highest in 2030. However, the power gain is still in a similar range. Comparing the red polar plots for P3 in Figures 6 and 7, the most significant difference is that the power gain for northeastern winds are lower while the power gain for bins with southern wind increases. Moreover, the 9 m/s-bins for western wind directions (255°and 285°) have higher power gains relative to the 7 and 11 m/s bins in the same sector. The power gain from P4 for the same bins in 2020 and 2030 is visually indistinguishable. The purple polar plots in Figures 6 and 7 show very similar colour patterns. However, there are less distinct differences in 2030 among the bins in the same sector with eastern directions. The maximum power gain attained slightly increases.

There are, however, differences between the participants. First, the maximum power gain ranges from 2% for P2 to 10% for P5. Participant P5 reports also the largest range of power gains. The low levels of power gain estimated by P2 might simply be related to the smaller layout with overall lower wake losses. In the larger layout considered by the rest of the participants, higher wake losses are observed which are expected to result in potentially higher WFFC gains. The polar plots for P2 and P4 for both years show local maxima for wind directions from the north (with bin centres of 15° and 345°), the east (with bin centres of 75° and 105°), the south (with bin centres of 165° and 195°) and the west (with a bin centre of 255° and 285°). These are sectors with high electricity prices or a high number of samples per bin (see Figure 1). While this applies for P4 with the full TC-RWP for two adjacent sectors, P2 with the subset of TC-RWP achieved this for one wind direction sector. The difference is more distinct for P2 than P4, where both P2 and P4 applied wake steering as WFFC strategy. This pattern of power gains across sectors does not appear for P3 and P5, which applied both wake steering and induction control to the TC-RWP layout. Having applied combined control strategies, their results show more variation between wind speed bins in the same sector. Instead of single or two adjacent sectors, advantageous wind directions seem to cover a broader range of wind





directions. An interesting observation is that P3 has the lowest and almost no power gain for western wind (sectors with 255°
and 285° in Figure 6), while P4 has the highest power gains for the same sectors. Although both P3 and P4 simulated the full
TC-RWP, the different control strategies applied are seen to be the main driver for that disparate behaviour.

## 4.3 Energy gain


Normalised Energy Gain during High Prices

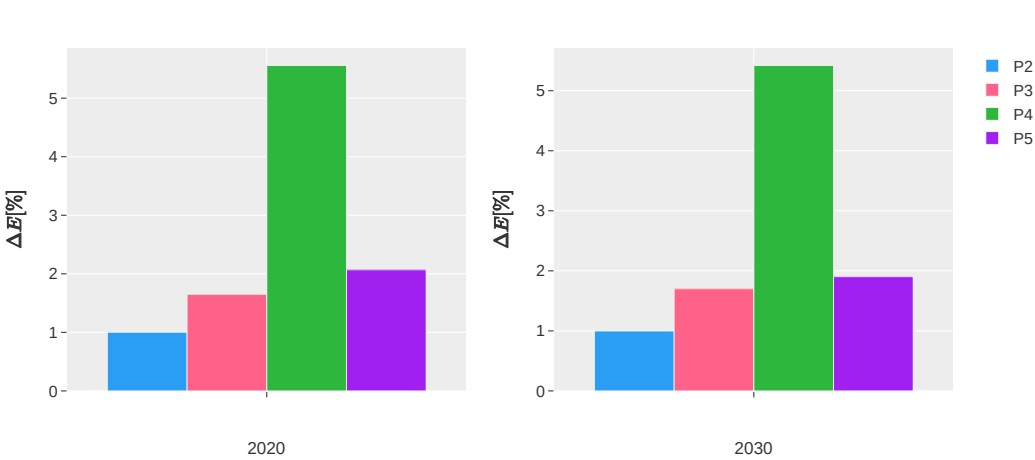

**Figure 8.** Energy gain ($\Delta E$) in percentage when using wind farm flow control (WFFC) in simulations with the wind inflow and price
information as shown in Figure 1 for 2020 (left) and 2030 (right), with respect to the production under normal operation without WFFC,
calculated using Equation 6 for each participant, and where normalisation is performed per bin. Each bar indicates the results for one of the
participants P2–P5.

Figure 8 summarises the energy gains during high prices for 2020 and 2030. Each bar represents the energy gain for one
participant, as normalised sum over all bins:

$$\Delta E_{\text{total}}^{\text{norm}} = \sum_{\text{bin}} \left[ \frac{E_{\text{WFFC}} - E_{\text{Normal Operation}}}{E_{\text{Normal Operation}}} \right]_{\text{bin}} \tag{6}$$

The generated energy $E_{\text{WFFC/Normal Operation}}$ [MWh] per bin is:

$$E_{\text{WFFC / Normal Operation}_{\text{bin}}} = n_{\text{bin}} \sum_{\text{WT}} P_{\text{WT}} \tag{7}$$


where $n_{\text{bin}}$ [h] is the number of samples in that bin, and $P_{\text{WT}}$ [MW] is the power produced at each wind turbine. In the showcases
dataset, bins have a one-hour resolution, such that multiplying samples and power production in a bin gives the energy.

The results for 2020 and 2030 are consistent for each participant reaching similar percentages in both years. Participant P2
achieves in both years a normalised energy gain of 1%, P3 of 1.7%, and P4 of 5.5%. The energy gain reported by P5 in 2020 is
2%, which is slightly higher than that in 2030 with 1.7%. This makes P5 the only participant with somewhat different energy






gains in 2020 and 2030. Participant P5 was also the one with the largest variance of power gains across the bins, reaching up to 10%. The normalised energy gains from P2, P3 and P5 range from 1% to 2%. Participant P4 reaching a much higher energy gain of 5% was the one with the most even power gains over all bins (see Figures 6 and 7). Furthermore, P3 and P5 have the same normalised energy gain in 2020 despite the large differences in power gains per sector.

All participants achieve a minimum of 1% energy gain. A recent expert elicitation revealed that wind farm operators and turbine manufacturers consider already an increase of the annual energy production of less than 1% as sufficient to justify the field implementation of WFFC (van Wingerden et al., 2020). This supports the general industrial potential of WFFC even without the consideration of the electricity prices.

## 4.4 Income gain

### Total Income Gain during High Prices

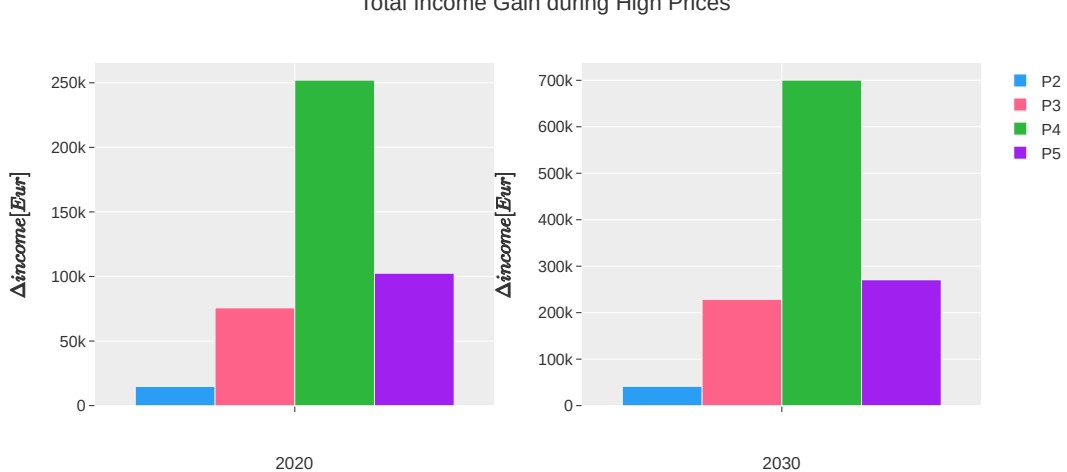

**Figure 9.** Total income gain ($\Delta income$) in EUR when using wind farm flow control (WFFC) in simulations with the wind inflow and price information as shown in Figure 1 for 2020 (left) and 2030 (right), compared to the production under normal operation without WFFC. Each bar indicates the results for one of the participants P2–P5.

The total income gain per participant during high prices for both years is shown in Figure 9. It is calculated by multiplying the energy gain per bin with the respective electricity price before summing it over all bins.

The income gain in 2030 is in general much higher than in 2020 because of the higher price level in that showcases set. Comparing the income gains per participant for both years, the numbers are consistent with the energy gains shown in Figure 8. As expected, P2 has the lowest income gains due to the smaller number of wind turbines in the simulated layout. Although all

the other participants used the TC-RWP, they report very different income gains as also the energy gains in Figure 8 suggest.

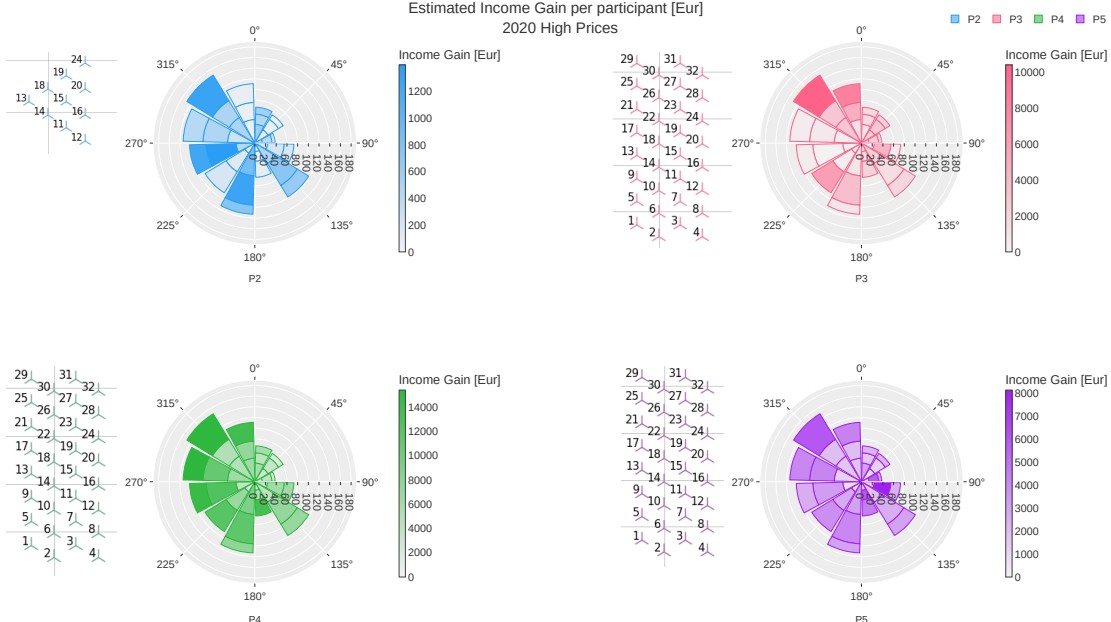

**Figure 10.** Income gain when using wind farm flow control (WFFC) in simulations with the wind inflow and price information for 2020 as shown in Figure 1. Each polar plot presents the results of one participant. The wind farm layout considered by this participant is shown on the left side of the polar plot.

The estimated income with WFFC is compared to the income under normal operation without WFFC. For each sector, the stacked gains are indicated as heat maps and reported per 7 m/s, 9 m/s and 11 m/s along the radial direction, with respect to the number of samples per bin.



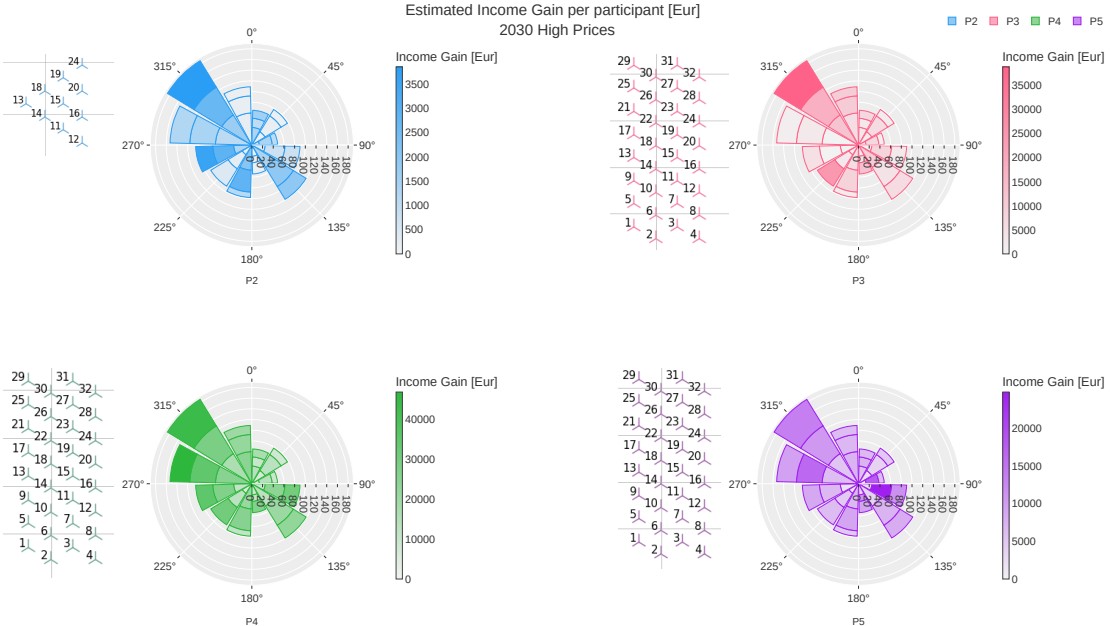

**Figure 11.** Income gain when using wind farm flow control (WFFC) in simulations with the wind inflow and price information for 2030 as shown in Figure 1. Each polar plot presents the results of one participant. The wind farm layout considered by this participant is shown on the left side of the polar plot.

The estimated income with WFFC is compared to the income under normal operation without WFFC. For each sector, the stacked gains are indicated as heat maps and reported per 7 m/s, 9 m/s and 11 m/s along the radial direction, with respect to the number of samples per bin.

The income gain per bin shown in Figures 10 and 11 is calculated as:

$$\text{Income Gain}_{\text{bin}} = \left[ E_{\text{WFFC}_{\text{bin}}} - E_{\text{Normal Operation}_{\text{bin}}} \right] \cdot \mathbb{P}_{\text{bin}} \tag{8}$$

where $\mathbb{P}_{\text{bin}}$ is the unit price [Eur/MWh] per bin. Accordingly, the income gain reported per participating model in Figures 10 and 11 during high prices are driven mainly by the power gain estimated as the variance in prices among the bins for the

investigated scenarios is low in Figure 1. Consequently for both 2020 and 2030 simulations, the sectors with estimated power gain are seen beneficial for the additional income via WFFC. However, the energy production is higher for the higher wind speed bins below rated and together with a larger number of samples per bin, the income gain for 9 and 11 m/s is expected to be higher. This is reflected by the darker colours observed at the outer rings of the polar plots in Figures 10 and 11.

Most significant across all participants and both years is the bin with a wind direction of 315°and a wind speed of 11 m/s,

which has a much darker shade in the income gains compared to the normalised power gains. In the 2020 scenario, there is also a distinct difference between the 11 m/s and the 9 m/s bin in this sector. Participant P3 has in 2020 high power gains for eastern wind sectors (45°- 115°). The income gain for these sectors is, however, rather low due to the low number of samples in these bins (illustrated by the radial axes in the polar plots). For P4, the dominant income gain for the 11 m/s-bins with western inflow (195°-345°) is very obvious while the 7 m/s-bins show lighter colours in Figures 10 and 11 compared to the normalised




power gain in Figures 6 and 7. In contrast to the other participants, P5 achieves also a high income gain for the 9 m/s bins in
the 75°- and 115°-sectors. The income gain is even higher than for the 11 m/s-315°-bin. This corresponds to the highest power
gains that P5 reports in the wind sectors with 75° and 115° inflow.

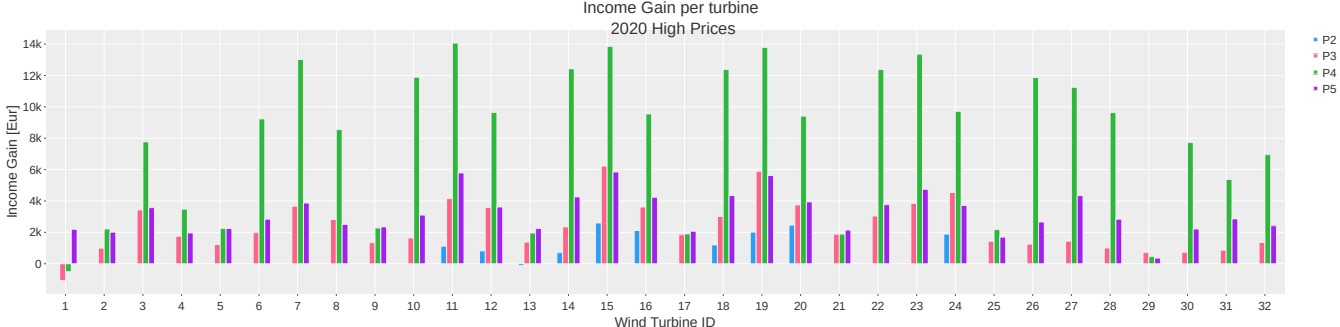

**Figure 12.** Income gain per wind turbine when using wind farm flow control (WFFC) in simulations with the wind inflow and price informa-
tion for 2020 as shown in Figure 1. Each bar shows the difference between the income with WFFC and the income under normal operation
without WFFC for one participant. The turbine IDs coincide with the numbering in the farm layout shown in e.g. Figure 11. Note that P2
investigates the subset layout and the results (in blue) are available for those 10 turbines only.

In order to investigate the contribution of the individual turbines to the total income gain, the reported values for 2020 and
2030 scenarios are broken down into the contribution of single wind turbines for the considered layouts in Figures 12 and
13. Each bar represents the income gain achieved by the turbine (with turbine IDs specified by the polar plots per participant
in, e.g., Figure 10), where the colour indicates the participants. The TC-RWP subset with 10 wind turbines considered by P2
includes WT11–WT16, WT18–WT20, and WT24.

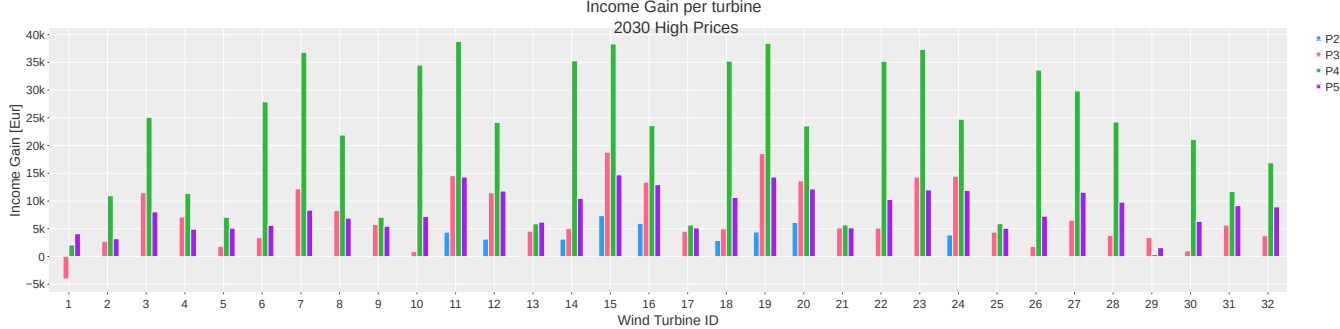

**Figure 13.** Income gain per wind turbine when using wind farm flow control (WFFC) in simulations with the wind inflow and price informa-
tion for 2030 as shown in Figure 1. Each bar shows the difference between the income with WFFC and the income under normal operation
without WFFC for one participant. The turbine IDs coincide with the numbering in the farm layout shown in e.g. Figure 11. Note that P2
investigates the subset layout and the results (in blue) are available for those 10 turbines only.



Although lower gains are reported due to the smaller layout investigated by P2, the normalised standard deviation (or the coefficient of variance) among the income gain per turbine reaches up to 60%, which is equivalent to P4 results with the 32-turbine layout. The highest variance is observed in P3 results, reaching more than 67% and the lowest is reported by P5 with less than 40%. This behaviour is the same for both 2020 and 2030, and it follows the trends presented in Figure 10 closely, where the sensitivity of the income gain to wind direction is observed to be the highest in P3 and lowest in P5 results for 2020 high electricity prices.

For the 2020 scenario, wind turbines with negative income gain are WT1 for P3 and P4 and WT13 for P2. These wind turbines are located in the upwind direction for inflow from the west where the bins contain many samples and relatively high prices (see Figure 1). The wind turbines with the highest income gains for all participants in 2020 are WT11, WT15 and WT19. These are located in the centre of the TC-RWP having the highest wake losses —thus the highest potential for wake mitigation via WFFC — from any wind direction. Conversely, turbine WT29 which is located at the northwest corner is reported to have very low income gain by P3, P4 and P5 with full layout investigation. This WT is the first upstream WT for wind coming from the dominant northwestern direction, such that it does not benefit from reduced wake losses due to WFFC but is rather sacrificed for the benefit of the whole wind farm. In the TC-RWP subset used by P2, the lower income gains for WT11 and WT19 can be explained by the reduced wake effect as they are located at the outside of the 10-turbine layout close to an edge.

As north-west winds become more frequent for high prices in 2030, the profile of the waked turbines changes slightly and only P3 reports an income loss for WT1 in Figure 13. The most beneficial turbines in terms of the income gain are the same as in 2020. However, slight differences in trends can be observed at the turbines located around the edge of the considered layout; e.g., WT11 for P2, WT2 for P4, and WT31 for P3 and P5, typically an increase compared to 2020 results.

During both 2020 and 2030 scenarios, it is very interesting to compare P3 and P5 results as both have implemented combined flow control strategies for their optimisation in full TC-RWP layout. Figures 12 and 13 show that the expected income gain provided by those two participants are indeed similar, especially for the turbines located in the centre of the wind farm. Therefore, it can be said that the differences observed for the turbines at the edge of the layout (upstream turbines) are the main drivers for the disparity in the overall income gain per wind direction sector presented in Figures 10 and 11. It also indicates potentially different control settings applied to those upstream turbines, resulting in dissimilar power losses. However, the overall benefit in terms of wake mitigation is quite comparable for the investigated high price scenarios in 2020 and 2030, as also shown in Figure 9.

## 5 Conclusions

This article presents the results of the FarmConners market showcases which are the first to study WFFC in simulations with variable electricity prices. The results from five participants are analysed and compared to demonstrate the potential benefit of WFFC in electricity market scenarios. The analysis starts at the individual turbine level with the examination of a method that applies different control strategies depending on the electricity price, and finishes at the wind farm level with a comparative study of four different implementations of WFFC strategies simulating scenarios with high electricity prices in 2020 and 2030.





All reported results are deterministic values for particular simulation environments. The uncertainties of the actual price signals and especially of wind forecasts can be in the same range as the reported gains here. While this is out of the scope of this conceptual study, an uncertainty quantification should be included in future investigations for a comprehensive evaluation of the estimated benefits per participating model and to identify the true value of the technology in the variable market scenarios.

For the former, further reading is encouraged on studies investigating the sensitivity and optimisation of widely used WFFC-oriented models under input uncertainties as well as the embedded uncertainties in the model parameters (Rott et al., 2018; Simley et al., 2020; Quick et al., 2020; Hulsman et al., 2020; van Beek et al., 2021; Howland, 2021). Accordingly, the present study should be read as an initial assessment of potential benefits when considering electricity prices in the operation of wind farms. With minimum 1% normalised energy gains for high prices reported by the participating models, it certainly motivates

further investigation of multi-objective WFFC.

The main outcomes and observations of the FarmConners market showcases are summarised in the list below. They are given as qualitative statements due to the associated uncertainties in the reported benefits and sorted according to the steps followed in the analysis.

**Benefit of flexible, market-driven, multi-objective control**  The benefit of revenue maximisation and structural load reduc-
tion as control objectives depending on the electricity prices is demonstrated at a single wind turbine. Extending this strategy to the operation of wind farms can be beneficial for all stakeholders, where the power production is increased when high demands lead to high electricity prices, and decreased when low demands cause low prices.

**Limited model capabilities to include structural load alleviation in the objective function**  The original FarmConners mar-
ket showcases defined three sets from which only the high-price scenarios could be analysed here. This is due to the lack
of participating models for the other sets which included constraining or reducing structural loads in the control objec-
tives.

**Impact of wind farm layout**  As already observed for WFFC power maximisation studies, the number of turbines in a wind farm and its layout clearly influences the achievable income gain per wind sector. The dominant wind direction and the upstream or downstream position of wind turbines has a significant impact on the operational strategy that is best suited
to reduce wake losses in a particular setting. The power gain per wind speed and direction bin then translates into income gain through the electricity prices distribution and relative frequency of occurrences.

**Consistent results for different participants**  The normalised gains of the four analysed WFFC implementations is consistent across the simulated cases. Despite the different control strategies and simulation set-ups, similar normalised energy gains are achieved. In particular, the two participants that apply wake steering show good agreement with respect to the
power gains in different wind sectors. The two participants that combine wake steering with axial induction control have more diverse results due to the larger range of potential control actions but lead to similar gains in total for the overall period.



**Benefit of maximising income instead of power gain** Maximising the income gain instead of the commonly maximised power gain has advantages. Unfavourable wind conditions in which the operational strategy only slightly increases the power gains can still result in high income gains if/when the electricity prices are high at the same time.

**Combination of factors in income gain** For particular meteorological conditions and layout, it has been showed how the specific combination of different factors, such as price distribution among wind speed and direction bins, their frequency of occurrence, available power for such bins (wind-speed dependent), wake losses reduction or WFFC strategy applied, makes the difference when it comes to income gain, rather than just the contribution of each factor independently. This combination could be integrated into the design of advanced WFFC strategies, beyond power maximisation. It also reflects how the time evolution of such combination can affect the economic assessment for a wind farm during its lifetime, e.g., if/when electricity market evolves.

*Code availability.* The notebooks for the market showcases results, including data snippets, can be achieved via the public repository of FarmConners Market Showcases (Göçmen et al., 2022).

*Data availability.* All the data used in FarmConners market showcases are available for non-commercial purposes. Please contact us using the details provided under the FarmConners market showcases wiki page https://farmconners-market-showcase.readthedocs.io/en/latest/contact_us.html.

*Author contributions.* KK, TG and IE organised the FarmConners market showcases, disseminated the showcases, prepared and shared the data sets, analysed the results and drafted most of the manuscript. LAAR, MAS, JF, JM, VP and IS participated in the FarmConners market showcases, prepared and ran the simulations and provided descriptions of their participating algorithms in sections 2 and 3. They are listed in alphabetical order of the last names.

*Competing interests.* JM is a member of the editorial board of Wind Energy Science. The peer-review process is guided by an independent editor, and the authors have also no other competing interests to declare.

*Acknowledgements.* The authors would like to thank Michael Smailes from ORE Catapult for his support on the organisation and dissemination of the FarmConners market showcases. Additionally, the authors thank Kaushik Das, Matti Juhani Koivisto, Juan Pablo Murcia Leon and Polyneikis Kanellas from DTU Wind Energy for the data generation by simulating the energy scenarios using the DTU Balancing Tool Chain and their contributions to the definition of the FarmConners Market Showcases.



The FarmConners market showcases are organised and conducted under the FarmConners project, funded by the European Union's Horizon 2020 research and innovation programme with grant agreement No 857844. The contribution by VP was funded nationally by

515   Stiftung Energieforschung Baden-Württemberg (FKZ: A341 21).



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
