# Peer review of "FarmConners Market Showcases Results: Wind farm flow control considering electricity prices and revenue"

_Wind Energy Science, 2022_

## Referee Comment (RC2)

**Article**: WES-2022-25

**Title:** FarmConners Market Showcases Results: Wind farm flow control considering electricity prices and revenue

**Authors:** Konstanze Kolle, Tuhfe Gocmen, Irene Eguinoa, Leonardo Andres Alcayaga Roman, Maria Aparicio-Sanchez, Ju Feng, Johan Meyers, Vasilis Pettas, and Ishaan Sood

**Overall comments**

The submitted article is a part of a series of papers related to the FarmConners project. Five different wind farm control methods (one individual turbine control, four wind farm flow control) are used to estimate the potential revenue increases in single 2020 and 2030 electricity price scenarios.

Overall, the motivation and contributions of the study could be more clearly explained. Results from different groups are analyzed but are not compared in detail to one another. If the focus isn't to compare the different strategies in detail, why are five distinct participants included? The stated focus of the article appears to be on evaluating methods for maximizing revenue, but the four flow control methods seem to maximize energy and the results are translated into energy after, using averaged electricity prices per wind direction sector. Also, since the authors focus on arbitrarily defined "high price" instances, it is hard to get a sense of the actual potential to increase yearly/farm lifetime revenue. Uncertainties (of which there are many) are discussed, which is great, but the discussion is limited to only one paragraph in the conclusions.

Overall, I believe this paper could have potential to provide insight into the market potential for wind farm control but I encourage the authors to consider the comments below, in the hope that they can help improve the paper. In particular, some decisions in the methods and discussions in the results seem qualitative or ad hoc, and could use more justification.

**General comments**

1.      I am not clear on why the authors have parsed the price timeseries to only consider "high prices" (arbitrarily set at the highest 25% of prices). The electricity prices don't have any impact on the power gain. But since the paper filters the wind conditions by the electricity prices for the different years, the wind conditions analyzed are different, since the electricity price timeseries are different between the two years considered. I would have thought that the "showcase" should focus on annually averaged metrics, or even better, farm lifetime averaged metrics (with uncertainty). I also think that looking at only one 2030 scenario of prices is pretty limiting, since there is substantial uncertainty around future energy prices.

2.      The study doesn't appear to actually do any revenue maximization, which I believe would have been much more interesting. The results demonstrated in this study take energy gain and multiply by wind direction sector averaged energy prices. But maximizing revenue (balancing short-term revenue

gains from power increases and long-term (potential) revenue gained/lost due to decreased/increase fatigue loads (causing changes in O&M, different lifetime, etc.)) could potentially lead to different control strategies, which would be interesting for the community.

**Point comments**

1.      Lines 2-4: *"For this, offshore wind will play a major role, significantly contributing to a paradigm shift in the power generation and greater volatility of electricity prices. The operating strategy of wind farms should therefore move from a power maximization to revenue maximisation design."*

I don't quite follow the logic in these two sentences and I would suggest rephrasing. Why would a greater volatility of electricity prices necessitate revenue maximization rather than power maximization? My initial thought after reading the first sentence was that farms should focus on system benefits (e.g. regulation services) rather and power maximization or revenue maximization.

2.      Line 12: "[…] and a favourable control strategy for dominant wind directions can pay off."

This statement is a bit vague. Do the authors mean that a revenue maximizing strategy differs from a power maximizing strategy? It would also help to include quantitative statements in the abstract.

3.      Section 1.2: I did not quite follow the strategy to determine energy price scenarios. It appears that 2020 and 2030 price scenario timeseries are generated. Then the price timeseries in parsed into low and high prices, corresponding to the lowest and highest 25% of the price data respectively? Please add a few more sentences explaining and justifying these selections. Why was no full year scenario run with all of the price data (rather than just 25% of the highest prices)?

4.      Line 83: Define power-boosting for individual turbine control.

5.      Line 94: Can the authors include more details and references for the P1 aeroelastic and surrogate modeling methodology?

6.      Line 98: What is meant by "desired trajectory"?

7.      Line 100: *"The whole process is automated so that given a desired trajectory, the relevant design variables can be estimated, i.e., the torque constant, rated values, cut in, and rated wind speed."*

I did not follow this sentence. What do the authors mean by estimating the relevant design variables? Are the authors considering revenue maximizing wind turbine design in this study? My understanding was that this study focused on wind farm flow control (WFFC), and there is an existing reference case with fixed turbine properties. Perhaps these variables need to be dynamically estimated because of the modified control? It's not clear to me why you would need to estimate these design variables if you have "power coefficient (Cp), tip speed ratio, and blade pitch angle" already.

8.    Figure 2: This is a helpful figure. The authors could emphasize that only turbine pitch is being modified to achieve the controller mode.

9.    Section 2: To ensure a self-contained article, the authors should briefly describe each controller mode shown in Figure 2.

10.   Line 118: How long does it take to achieve steady-state behavior in the FAST simulations? Is this hysteresis neglected in the timeseries evaluation of the controller?

11.   Figure 3:

    a.    Perhaps make it clear in the caption that some figures show a subset of the 2030 timeseries while others show the full timeseries.
    b.    Bottom left subfigure: Should the y-axis be "instantaneous power" instead of "rated power"?
    c.    Bottom right subfigure: This plot could use more explanation. Why is there a large initial transient in each case? I also find the terminology "Accumulated DEL" confusing when it is being computed using Eq. (1).

12.   Line 166: Can the authors provide more details of the tuning process to give insight into challenges and research gaps? What input variables are the authors setting user-defined thresholds on? Just the PI pitch controller?

13.   Line 170: This paragraph's discussion and the results in Figure 5 are very interesting! It would be great if the authors could provide more discussion about challenges and opportunities for multi-objective optimization of revenue and loads based on your results (as also related to the previous comment). It appears from the discussion that the empirical results shown here depend strongly on the tuning.

14.   Figure 5 is very small, consider increasing the figures and the fontsize.

15.   Line 175: This statement makes it more clear why the details were very limited here. No reference is provided for this method, just an indication of a future study. This is not ideal, as articles should be self-contained, or at a minimum have adequate descriptions and references to peer-reviewed publications which are accessible to readers.

16.   Table 2:

    a.    I recognize 'Gauss-legacy' is the FLORIS terminology, and useful to include, but can the authors explain in the table caption why it is called 'legacy' to inform readers who aren't as familiar with FLORIS.
    b.    P3: Citation missing for linear superposition. Is this linear superposition of deficits with respect to freestream or the local velocity [1]?
    c.    P2 and P4 are using identical model setups?

17.   Table 3:

> a. Add a definition of each parameter introduced in this table. It would be much easier for readers to make this paper more self-contained, rather than having to refer to FLORIS documentation (which is also not archival since GitHub repositories can evolve).
>
> b. Why are different values of n used by P2 and P4? These values of n depend on the turbine model, the wind speed and direction shear [2], and the waked conditions [3]. Since the test case is the same between the participants (with DTU-10MW turbines), I would have expected n to be the same.

18. Section 3.3: How are the wind shear exponents found? Please include more details. To which data are the power-law curves fit? Please also explain how the wind shear exponent is used by the wake models in FLORIS.

19. Section 3.4: What is the power-yaw exponent for P5?

20. Figure 6: I suggest having the same colorbar axis limits for all of the subfigures.

21. Line 346: *"Adjacent wind speed bins with high and low power gains in the same sector are observed exclusively in P5 results."*

This also appears to happen with P3 (e.g. northerly flow), or am I misunderstanding this statement?

22. Line 348: *"The discussed Figures 6 and 7 show consistent power gains per participant for both years."*

What do the authors mean by "consistent"? Consistent between 2020 and 2030 or consistent between the different participants. If it is the latter, I would suggest the results have significant spread.

23. Line 373: *"Although both P3 and P4 simulated the full TC-RWP, the different control strategies applied are seen to be the main driver for that disparate behaviour."*

Can we definitively say this or can it also relate to the different wind farm model?

24. Line 384: *"The energy gain reported by P5 in 2020 is 2%, which is slightly higher than that in 2030 with 1.7%."*

Are these differences (between 1.7% and 2%) statistically significant?

25. Figure 8:
> a. Is the high energy gain associated with P4 due to the discretization of the wind directions within a wind direction bin?
>
> b. With this (and other bulk metric figures) it probably makes sense to plot the income gain per turbine so that P2 isn't falsely interpreted as an outlier.

26. Line 395: How is income gain computed? Is the power increase for each time step multiplied by the cost of electricity within that timestep? In the introduction, the study is motivated by the time-varying nature of the energy prices, so it would seem natural to do a timeseries analysis rather than using a mean energy price per wind direction sector.

27. Line 403: Sentence starting "Accordingly […]" is a bit confusing. I didn't understand the point being made. Consider rephrasing.

28. Line 469: *"The benefit of revenue maximisation and structural load reduction as control objectives depending on the electricity prices is demonstrated at a single wind turbine"*

As in previous comments, I don't think the authors have actually done revenue maximization. They have done power maximization and then translated the results into economic terms through a price for energy. I expect that a revenue maximization approach would focus on balancing short-term revenue gains from power increases and long-term (potential) revenue gained/lost due to decreased/increase fatigue loads (causing changes in O&M, different lifetime, etc.).

29. Line 482: *"The normalised gains of the four analysed WFFC implementations is consistent across the simulated cases."*

Not sure I agree with this qualitative statement that the different participants have "consistent" results but since it is qualitative it is subjective. It appears that the results differ in some cases by a factor of two or more, which seems significant.

30. Line 488: *"Benefit of maximising income instead of power gain"*

Which results in this study demonstrate this conclusion?

**References**

[1] Niayifar, Amin, and Fernando Porté-Agel. "Analytical modeling of wind farms: A new approach for power prediction." *Energies* 9, no. 9 (2016): 741.

[2] Howland, Michael F., Carlos Moral González, Juan José Pena Martínez, Jesús Bas Quesada, Felipe Palou Larranaga, Neeraj K. Yadav, Jasvipul S. Chawla, and John O. Dabiri. "Influence of atmospheric conditions on the power production of utility-scale wind turbines in yaw misalignment." *Journal of Renewable and Sustainable Energy* 12, no. 6 (2020): 063307.

[3] Liew, Jaime, Albert M. Urbán, and Søren Juhl Andersen. "Analytical model for the power–yaw sensitivity of wind turbines operating in full wake." *Wind Energy Science* 5, no. 1 (2020): 427-437.

---

## Author Comment (AC1)

**Response to the reviewers**

Konstanze Kölle *et al.*

10 August 2022

**Reviewer 1**

*Comments to the Author:*

This paper presents an interesting study wherein different participants model the benefits of turbine or farm control in a market context and compare the impact both on energy production as well as expected revenue. I think it is a really useful exercise, to see these comparisons made in an open-access journal, and using separate research departments, models and controllers adds a degree of internal variation that is really helpful in assessing how particular the results are to any specific set of assumptions.

Reading the paper I didn't identify too much to critisize. I therefore have just a general comments.

*Comment No. 1:*
My main thought was that some of the comparisons might benefit from some extra context. In the case of energy gain, I was interested to see (and apologies if I simply missed it) some information on how each model predicts the overall starting wake losses for the two farm types. In other words the percentage difference in AEP for the farm with and without wakes.

*Response to comment No. 1:* We will provide a measure of wake loss utilising the available results from the participants for the simulated showcase. The simulations were performed with and without wind farm flow control. The latter provides the predicted power output with each wake model which can be related to the theoretical power output without wakes using the turbine's power curve and the inflow wind speed.

However, the data set does not cover a whole year and the wake loss will only be computed for the high-prices showcases.

*Comment No. 2:*
Then for additioinal context on the revenue side, what would be the overall initial expected revenue of the farm. What I would be very curious to see if it would be possible would be table where each case study (P2-P5) is a row with

columns like: annual revenue, delta revenue, annual profit (if it were possible to assume some cost model), expected delta profit. Was just wanted to see a rough gauge on the significance in these changes in revenue.

*Response to comment No. 2:*
We agree that the annual revenue and profit are very interesting measures. However, for the current study, the available data from P2-P5 cover only a fraction of a year due to the design of the showcases which was decided earlier. As we cannot provide annual metrics here, we have to leave those to future studies. To be more specific in the present article, we suggest to rename it into *'FarmConners Market Showcases: Wind farm flow control considering electricity prices'*, i.e., removing the 'and revenue' from the title. Likewise, we will review all occurrences of the term 'revenue' to make sure it is used consistently.

*Comment No. 3:*
Finally, I had a question on comparing the EUR in 2020 versus 2030. I'm not an expert in economics, but was wondering, is there a need to control the value of the EUR between these 10 years, so that both are expressed in 2020 EUR or similar?

*Response to comment No. 3:*
The prices are actually in € 2012 value complying with the considered meteorological data from 2012. We will include a clarifying comment about 2012 being the reference year for the electricity price in the revised version. The energy scenarios including e.g. radial offshore connections in 2030, however, differentiate 2020 from 2030.

**Reviewer 2**

**Overall comments**

The submitted article is a part of a series of papers related to the FarmConners project. Five different wind farm control methods (one individual turbine control, four wind farm flow control) are used to estimate the potential revenue increases in single 2020 and 2030 electricity price scenarios.

Overall, the motivation and contributions of the study could be more clearly explained. Results from different groups are analyzed but are not compared in detail to one another. If the focus isn't to compare the different strategies in detail, why are five distinct participants included? The stated focus of the article appears to be on evaluating methods for maximizing revenue, but the four flow control methods seem to maximize energy and the results are translated into energy after, using averaged electricity prices per wind direction sector. Also, since the authors focus on arbitrarily defined "high price" instances, it is hard to get a sense of the actual potential to increase yearly/farm lifetime revenue.

Uncertainties (of which there are many) are discussed, which is great, but the discussion is limited to only one paragraph in the conclusions.

Overall, I believe this paper could have potential to provide insight into the market potential for wind farm control but I encourage the authors to consider the comments below, in the hope that they can help improve the paper. In particular, some decisions in the methods and discussions in the results seem qualitative or ad hoc, and could use more justification.

*Response to overall comments:*
The study is the first that includes electricity prices in the evaluation of wind farm flow control. A higher share of renewable energy sources in the energy mix will provide additional incentives beyond power maximisation. Wind farms could, for example, participate in balancing markets and provide frequency support to the power system.

There are uncertainties related to the general benefit of wind farm flow control. The differences in academic implementations alone include various flow models, different control strategies and algorithms, and tuning procedures [1]. Including several participants with their different codes represents this variability. However, a comparison of farm flow models is provided, e.g., in [1].

In summary, the motivation of the current study is

- to estimate the general benefit that market-driven wind farm flow control potentially has,

- to investigate the readiness or state of the art of wind farm flow control for participation in electricity markets,

- reveal the gaps of market-driven wind farm control including farm-wide flow phenomena.

**General comments**

*Comment No. 1:*
I am not clear on why the authors have parsed the price timeseries to only consider "high prices" (arbitrarily set at the highest 25% of prices). The electricity prices don't have any impact on the power gain. But since the paper filters the wind conditions by the electricity prices for the different years, the wind conditions analyzed are different, since the electricity price timeseries are different between the two years considered. I would have thought that the "showcase" should focus on annually averaged metrics, or even better, farm lifetime averaged metrics (with uncertainty). I also think that looking at only one 2030 scenario of prices is pretty limiting, since there is substantial uncertainty around future energy prices.

*Response to comment No. 1:*

| Showcases set | Performance indicators |
|---|---|
| 1) High prices | income gain |
| 2) Low prices | income gain (+ load alleviation as restriction) |
| 3) TSO-driven | load alleviation, index for tracking of reference from TSO |

Table 1: Performance indicators per showcase set. TSO : transmission system operator

The showcases are built from a time series of a whole year. Originally, three showcases sets were defined as shown in Table 1. The one-year time series was split into these three showcases sets to provide incentives to not maximise power at any time. Without additional constraints, power maximisation would be the primary objective throughout the whole year. The *2) Low prices* and *3) TSO-driven* cases have performance indicators different from income gain only.

It was up to the participants to decide in which showcases they wanted to participate and how they would perform the simulations. All participants chose showcase *1) High prices*, and to maximise power in order to increase the income. There were no participants for the other two showcases.

This shows not least the current state of the research field focusing on power maximisation. In the revised version, we will elaborate on the background of the showcases sets and the choice by the participants to emphasise the conclusion that the majority of research codes for wind farm control is not ready to balance several objectives.

We agree that considering annual or lifetime metrics will be important in the long term to evaluate revenue-maximising strategies. However, it is not directly beneficial for the current study which – despite the discussed limitations – is the first to analyse electricity prices in the evaluation of wind farm flow control starting with two energy scenarios. Future studies should investigate the impact of uncertain electricity prices in various future scenarios.

*Comment No. 2:*
The study doesn't appear to actually do any revenue maximization, which I believe would have been much more interesting. The results demonstrated in this study take energy gain and multiply by wind direction sector averaged energy prices. But maximizing revenue (balancing short-term revenue gains from power increases and long-term (potential) revenue gained/lost due to decreased/increase fatigue loads (causing changes in O&M, different lifetime, etc.)) could potentially lead to different control strategies, which would be interesting for the community.

*Response to comment No. 2:*
This statement is partly true. Revenue maximisation has been done for the single wind turbine whereas it was not maximised for a wind farm, *per se*.

We agree that the outlined revenue maximisation balancing short- and longterm gains would be very interesting and of great use for the research community. However, such analysis requires much longer time-series simulations. This is, as also discussed in the reply to general comment No. 1, not within the scope here.

To meet the criticism, we suggest to rename the article into *'FarmConners Market Showcases: Wind farm flow control considering electricity prices'*, i.e., removing the 'and revenue' from the title.

**Point comments**

*Comment No. 1:*
Lines 2-4: "For this, offshore wind will play a major role, significantly contributing to a paradigm shift in the power generation and greater volatility of electricity prices. The operating strategy of wind farms should therefore move from a power maximization to revenue maximisation design."

I don't quite follow the logic in these two sentences and I would suggest rephrasing. Why would a greater volatility of electricity prices necessitate revenue maximization rather than power maximization? My initial thought after reading the first sentence was that farms should focus on system benefits (e.g. regulation services) rather and power maximization or revenue maximization.

*Response to comment No. 1:*
In case of low electricity prices, it could be beneficial for the wind farm operator to produce less power than available. Such operation can contribute to the power reserve and it is possible to choose operational points where less fatigue is accumulated reducing O&M costs on the long run. The turbines can be rather 'sacrificed' in situations with high prices when more income can be generated.

We suggest the rephrased sentences: "For this, offshore wind will play a major role, significantly contributing to a paradigm shift in the power generation and greater volatility of electricity prices. The operating strategy of wind farms should therefore move from power maximisation to profit maximisation which includes income from providing power system services and the reduction of maintenance costs."

*Comment No. 2:*
Line 12: "[. . . ] and a favourable control strategy for dominant wind directions can pay off."

This statement is a bit vague. Do the authors mean that a revenue maximizing strategy differs from a power maximizing strategy? It would also help to include quantitative statements in the abstract.

*Response to comment No. 2:*
A large power gain for dominant wind directions will pay off even if the electricity prices are low. We suggest to change the sentence to: "[...] and a favourable control strategy for dominant wind directions can pay off even for low electricity

prices".

*Comment No. 3:*
Section 1.2: I did not quite follow the strategy to determine energy price scenarios. It appears that 2020 and 2030 price scenario timeseries are generated. Then the price timeseries in parsed into low and high prices, corresponding to the lowest and highest 25% of the price data respectively? Please add a few more sentences explaining and justifying these selections. Why was no full year scenario run with all of the price data (rather than just 25% of the highest prices)?

*Response to comment No. 3:*
The division into three showcases sets with different control objectives should artificially introduce incentives for the wind farm operator to not maximise the power at any time.

We will explain the idea of the original showcases sets in the revised paper with more details. Please see also our response to the general comment No. 1.

Please note that Participant 1 used the whole one-year time series for the simulations of a single wind turbine.

*Comment No. 4:*
Line 83: Define power-boosting for individual turbine control.

*Response to comment No. 4:*
Power boosting for the individual turbine is based on changing the controller's set points in terms of blade pitch, generator torque and/or generator speed effectively changing the power output. This is already offered by some manufacturers as explained from line 87. In this case, the power boosting is performed by increasing the rated torque and keeping the rated speed set-point constant as described in lines 107-110. We will add a reference to the OEM product and add some more explanations in line 83 to make it clearer.

*Comment No. 5:*
Line 94: Can the authors include more details and references for the P1 aeroelastic and surrogate modeling methodology?

*Response to comment No. 5:*
The sampling of the variable space and the aeroelastic simulation set-up is explained in lines 112-121. These include the aeroelastic code used, the conditions (wind speed, TI and shear), and the duration of the simulations. The interpolation and smoothening process for the surrogate model are explained in lines 123-127. We believe that this information ensures the reproducibility of the presented results. However, for clarity we will mention that FAST was used as aeroelastic code earlier in this paragraph.

*Comment No. 6:*
Line 98: What is meant by "desired trajectory"?

*Response to comment No. 6:*

The desired trajectory refers to the chosen set points assigned to the controller in order to achieve the different power output values. It is the trajectory followed within the $C_p - \lambda - \theta$ maps described in line 99-101. There are multiple combinations of tip speed ratio and pitch angle that can produce the same $C_p$ output with different trade-offs in terms of loads (and power in control region 1.5). We will rephrase the text starting from line 98 to: "(...) and power-boost modes. This is achieved by changing the set-points for power coefficient ($C_p$), tip speed ratio ($\lambda$) and blade pitch angles ($\theta$). The desired set-point trajectories for the modes are identified from $C_p - \lambda - \theta$-maps, which are (...)"

*Comment No. 7:*

Line 100: "The whole process is automated so that given a desired trajectory, the relevant design variables can be estimated, i.e., the torque constant, rated values, cut in, and rated wind speed."

I did not follow this sentence. What do the authors mean by estimating the relevant design variables? Are the authors considering revenue maximizing wind turbine design in this study? My understanding was that this study focused on wind farm flow control (WFFC), and there is an existing reference case with fixed turbine properties. Perhaps these variables need to be dynamically estimated because of the modified control? It's not clear to me why you would need to estimate these design variables if you have "power coefficient (Cp), tip speed ratio, and blade pitch angle" already.

*Response to comment No. 7:*

We will rephrase the sentence to make it clearer for the reader. As the power output changes, the controller parameters change including:

- the torque constant utilized in control region 2

- the wind speed at which the turbine reaches rated power

- the targeted set point for the rated torque value,

- the rated rotor speed and torque

- at low wind speeds -due to limits in possible reduction of the minimum rotor speed due to crossing with the tower frequencies- the down-regulation cannot be followed up to the cut in speed but has to start later depending on the set points

These variables are required for the controller setup and the optimization process. Since the main focus of the paper is not on controller tuning, we believe that going into too much detail here will be counter-productive in the context of the paper's scope.

The turbine design properties are not changed in any way, it is only a software adaptation in the controller. As explained in lines 83-91, power boosting,

down-regulation and IPC are existing technologies already offered by the industry for different purposes. The novel approach is to combine existing methods for optimizing the total revenue and fatigue accumulation by changing the operation modes of the turbine taking into account the electricity prices. We will update the text to make this clearer.

*Comment No. 8:*
Figure 2: This is a helpful figure. The authors could emphasize that only turbine pitch is being modified to achieve the controller mode.

*Response to comment No. 8:*
Thanks! Besides the pitch angles, the torque controller set points have to be changed, too. For example, to achieve down-regulation in below rated conditions we need to adjust the torque constant according to the desired power coefficient. Moreover, figure 2 explains the general methodology and could be achieved with any other trajectory achieving the same $C_p$ with other possible trade-offs in structural loading and actuator usage. The adjustments related to the previous comment will support a better understanding for the reader.

*Comment No. 9:*
Section 2: To ensure a self-contained article, the authors should briefly describe each controller mode shown in Figure 2.

*Response to comment No. 9:*
The methodology to adjust set points for down-regulation and power boosting is described in lines 93-115. The concept of changing set points has been discussed in the literature, with relevant studies already cited (Astrain Juangarcia et al., 2018; D.C. van der Hoek; and Kanev, 2017). The IBC loop is decoupled from the collective pitch controller and is explained in detail in the cited work (Pettas and Cheng 2018). From the authors' point of view, a more detailed discussion would be out of the scope of this work.

*Comment No. 10:*
Line 118: How long does it take to achieve steady-state behavior in the FAST simulations? Is this hysteresis neglected in the timeseries evaluation of the controller?

*Response to comment No. 10:*
The simulations described in line 118 are turbulent and therefore do not achieve steady behaviour. As described in the text the duration is one hour. In case the reviewer refers to the possible initial transients of the simulations, this is taken care of by using the circular wind fields created by TurbSim which allow the same wind field to be repeated, leading to the same statistical properties when a part is removed. The duration of the wind field was 3600 s and the simulation time was 3700 s with the first 100 s removed.

*Comment No. 11:*

Figure 3:

a. Perhaps make it clear in the caption that some figures show a subset of the 2030 timeseries while others show the full timeseries.

b. Bottom left subfigure: Should the y-axis be "instantaneous power" instead of "rated power"?

c. Bottom right subfigure: This plot could use more explanation. Why is there a large initial transient in each case? I also find the terminology "Accumulated DEL" confusing when it is being computed using Eq. (1).

*Response to comment No. 11:*

a. We will add the clarification to the caption as suggested by the reviewer

b. This is not the instantaneous power of the turbine but the rating that shows whether we are in down-regulation or power boosting mode. E.g. in 9 m/s wind speed when the rating is 9 MW the turbine produces 90% of the power compared to the baseline. We will make it more clear by replacing the y-label with Power Output in percentage (where 100% corresponded to a 10 MW rated power)

c. The transient, in the beginning, comes because we start with 0 DEL. In the beginning, every small change contributes more to the DEL according to the number of events (see eq 1). As the accumulation continues the weight of each added DEL is smaller making the rate of change slower in time. Moreover, at the beginning of the calendar year (January), wind speeds are generally higher due to the winter season, the DELs are in general higher than in the following months leading to this temporary overshoot of the accumulated value. We will change the y-axis label to cumulative DEL to match equation 1.

*Comment No. 12:*
Line 166: Can the authors provide more details of the tuning process to give insight into challenges and research gaps? What input variables are the authors setting user-defined thresholds on? Just the PI pitch controller?

*Response to comment No. 12:*
The tuning process involves parameters such as the amount of price/wind speed bins used, low-revenue threshold to shut down the turbine, upper revenue threshold to go into maximum full boost, etc. In the optimization itself, the weights of the objective function and the penalization of values had to be tuned differently for each of the optimization objectives shown. For this initial application, this was done manually by trial and error and a methodology to cover how this can be automated is not implemented.

The PI pitch controller is neither input nor output variable in the optimization process. The inputs are the price/speed values and the surrogates. The

output is the power output level to be used per wind speed-price bin to achieve cumulative DEL and revenue objectives.

Since this paper does not focus on the analysis of this method and these are initial results to demonstrate the effectiveness, we believe it is not relevant for the community and the scope of the paper to go into such details here. By showing an initial and simple application of the suggested process we want to demonstrate that flexible turbine operation, considering electricity prices as input and revenue and fatigue as objectives, can lead to improvements in long-term objectives.

More insight on the detailed parameters, challenges and research gaps will be included in a dedicated publication that is currently under preparation.

*Comment No. 13:*
Line 170: This paragraph's discussion and the results in Figure 5 are very interesting! It would be great if the authors could provide more discussion about challenges and opportunities for multi-objective optimization of revenue and loads based on your results (as also related to the previous comment). It appears from the discussion that the empirical results shown here depend strongly on the tuning.

*Response to comment No. 13:* As in our previous answer, we agree with the reviewer that this discussion is interesting. Nevertheless, we believe that this discussion would have limited benefit here since the current paper has a different scope.

*Comment No. 14:*
Figure 5 is very small, consider increasing the figures and the fontsize.

*Response to comment No. 14:* The figure will be adjusted according to the reviewer's recommendation.

*Comment No. 15:*
Line 175: This statement makes it more clear why the details were very limited here. No reference is provided for this method, just an indication of a future study. This is not ideal, as articles should be self-contained, or at a minimum have adequate descriptions and references to peer-reviewed publications which are accessible to readers.

*Response to comment No. 15:* We will add the presentation from the WESC 2021 conference (zenodo.org/record/5017956) to the references, where this method was initially introduced. Another manuscript is currently under preparation focusing exclusively on the single turbine optimization of revenue and loads including controller tuning, surrogate modelling and optimization approaches with more realistic preview horizons. In case this is finished until the publication of the current work it will be included as well. As this is a novel field of research there are no other references we are aware of. Nevertheless, we believe

that the descriptions included in the manuscript are adequate for the reader to understand the general methodology. The controller design and surrogate model approach are explained and combined with references. Their combination and application to the simulation and optimization framework are explained and illustrated. The optimizers inputs for this study are based on trial and error, a method which hardly can be further elaborated.

*Comment No. 16:*
Table 2:

    a. I recognize 'Gauss-legacy' is the FLORIS terminology, and useful to include, but can the authors explain in the table caption why it is called 'legacy' to inform readers who aren't as familiar with FLORIS.

    b. P3: Citation missing for linear superposition. Is this linear superposition of deficits with respect to freestream or the local velocity [1]?

    c. P2 and P4 are using identical model setups?

*Response to comment No. 16:*

    a. Thanks for the suggestion, the following comment is now included in the table caption:

    *The term Gauss-legacy is used in FLORIS Version 2 as opposed to the Gaussian formulation implementing the near wake model by (Blondel and Cathelain,2020), as described in (Fleming et al., 2020).*

    b. Linear superposition is intrinsic to Fuga [3] because it is a linearized solution to a non-linear forcing. PyWake superimposes deficits calculated for each individual turbine, respect to the local free stream, which takes into account any wake deficit from upstream turbines, if any. The reference will be added to the manuscript.

    c. The underlying models are the same (Table 2), but the parameters tuning is different, as can be seen in Table 3. P4 has used the default setting of FLORIS, with default parameters in the FLORIS code [4].

*Comment No. 17:*
Table 3:

    a. Add a definition of each parameter introduced in this table. It would be much easier for readers to make this paper more self-contained, rather than having to refer to FLORIS documentation (which is also not archival since GitHub repositories can evolve).

    b. Why are different values of n used by P2 and P4? These values of n depend on the turbine model, the wind speed and direction shear [2], and the waked conditions [3]. Since the test case is the same between the participants (with DTU-10MW turbines), I would have expected n to be the same.

*Response to comment No. 17:*

a. We will include this.

b. P2 and P4 used different procedures for tuning. P4 has used the default value $n = 1.88$ in FLORIS, which was determined based on high fidelity CFD simulations by Gebraad et al. in [5].

*Comment No. 18:*
Section 3.3: How are the wind shear exponents found? Please include more details. To which data are the power-law curves fit? Please also explain how the wind shear exponent is used by the wake models in FLORIS.

*Response to comment No. 18:*
In P4's simulation, the wind shear exponent was derived by fitting the power-law curve with mean wind speeds at two heights: 50 m and 150 m. This also means that the wind shear exponent for each flow case is derived independently based on the provided measurement data. This has already been described in section 3.3.1 by:

*Note that for simulating yawed and non-yawed wind farm production for each inflow bin, the wind shear exponent was derived with a power law based on the mean wind speed at two heights: 50 m and 150 m,*

In FLORIS, the wind shear exponent is used to model the 3D background flow field with regards to a specific hub height wind speed. Details can be found in the implementation of the *\_compute\_initialized\_domain* function of the Flow-Field class in the Python script *flow\_field.py* of the FLORIS code [4].

*Comment No. 19:*
Section 3.4: What is the power-yaw exponent for P5?

*Response to comment No. 19:*
As mentioned in section 3.4.2, the participant P5 uses the same expression for turbines under yaw as [2] for the DTU 10MW. The BEM derived expression for power coefficient under yawed conditions, i.e. $C_P(\gamma) = C_P * cos^3\gamma$, is multiplied by a scaling factor $\eta(\gamma) = \frac{1.08}{cos\gamma}$.

*Comment No. 20:*
Figure 6: I suggest having the same colorbar axis limits for all of the subfigures.

*Response to comment No. 20:* We considered this when preparing the draft. It was very difficult to see the results for P2 with the axis limits [0; 10]. However, we will re-consider it for the revision.

*Comment No. 21:*
Line 346: "Adjacent wind speed bins with high and low power gains in the same sector are observed exclusively in P5 results."

This also appears to happen with P3 (e.g. northerly flow), or am I misunderstanding this statement?

*Response to comment No. 21:*
The same observation can be made for P3. However, the difference observed for adjacent bins ($\sim$2% vs. $\sim$10%) is larger for P5. This may be better visible with the same axis limits as suggested in point comment No. 20. We will make the statement on Line 346 clearer by introducing values.

*Comment No. 22:*
Line 348: "The discussed Figures 6 and 7 show consistent power gains per participant for both years."

What do the authors mean by "consistent"? Consistent between 2020 and 2030 or consistent between the different participants. If it is the latter, I would suggest the results have significant spread.

*Response to comment No. 22:*
The intention was to describe the consistency between the two years. The sentence will be rephrased to: "The discussed Figures 6 and 7 show consistent power gains between 2020 and 2030 for each participant."

*Comment No. 23:*
Line 373: "Although both P3 and P4 simulated the full TC-RWP, the different control strategies applied are seen to be the main driver for that disparate behaviour."

Can we definitively say this or can it also relate to the different wind farm model?

*Response to comment No. 23:*
We cannot exclude this and will therefore rephrase the statement to something like: "Although both P3 and P4 simulated the full TC-RWP, the different control strategies applied are considered to be the main driver for that disparate behaviour even though the different flow models will also have an impact."

*Comment No. 24:*
Line 384: "The energy gain reported by P5 in 2020 is 2%, which is slightly higher than that in 2030 with 1.7%."

Are these differences (between 1.7% and 2%) statistically significant?

*Response to comment No. 24:*
Given the other uncertainties, this is not statistically significant. We will remove the values and write: "The energy gain reported by P5 in 2020 is about the same as that in 2030."

*Comment No. 25:*
Figure 8:

a. Is the high energy gain associated with P4 due to the discretization of the wind directions within a wind direction bin?

b. With this (and other bulk metric figures) it probably makes sense to plot the income gain per turbine so that P2 isn't falsely interpreted as an outlier.

*Response to comment No. 25:*

a. Yes, the discretization method taken by P4 tends to yield a more optimistic energy gain, since the yaw angles are optimized for each degree inside the wind direction bins. This has been discussed in the last paragraph of section 3.3.2, as:

*By considering 31 flow cases with different wind directions and solving the yaw optimisation problem separately for each inflow bin, P4 took an idealised or 'greedy' approach that tends to explore the full potential of wake steering, since the effectiveness of wake steering can be quite sensitive to the wind direction. However, in real-life implementation, limits on the speed and accuracy of the yawing system, uncertainty of the measured inflow wind direction, and other factors can make the reported energy gain hard to be fully realised.*

b. Good idea! We will consider that.

*Comment No. 26:*
Line 395: How is income gain computed? Is the power increase for each time step multiplied by the cost of electricity within that timestep? In the introduction, the study is motivated by the time-varying nature of the energy prices, so it would seem natural to do a timeseries analysis rather than using a mean energy price per wind direction sector.

*Response to comment No. 26:*
Yes, the power increase per bin is multiplied with the electricity price. Please see also the response to general comment No. 1. The showcases sets (including the one with high prices) were defined to analyse the impact of electricity prices on the outcome using wind farm flow control. A pure time series analysis was not feasible in this context.

*Comment No. 27:*
Line 403: Sentence starting "Accordingly [...]" is a bit confusing. I didn't understand the point being made. Consider rephrasing.

*Response to comment No. 27:*
Okay, will be changed to: "The variability in prices among the bins is low for the investigated high-prices showcase as shown in Figure 1. Accordingly, the income gain in Figures 10 and 11 reported per participating model during the high prices is mainly driven by the estimated power gain."

*Comment No. 28:*

Line 469: "The benefit of revenue maximisation and structural load reduction as control objectives depending on the electricity prices is demonstrated at a single wind turbine"

As in previous comments, I don't think the authors have actually done revenue maximization. They have done power maximization and then translated the results into economic terms through a price for energy. I expect that a revenue maximization approach would focus on balancing short-term revenue gains from power increases and long-term (potential) revenue gained/lost due to decreased/increase fatigue loads (causing changes in O&M, different lifetime, etc.).

*Response to comment No. 28:*
As stated in the previous comments you refer to, we will change the wording for a consistent use of the term "revenue".

*Comment No. 29:*

Line 482: "The normalised gains of the four analysed WFFC implementations is consistent across the simulated cases."

Not sure I agree with this qualitative statement that the different participants have "consistent" results but since it is qualitative it is subjective. It appears that the results differ in some cases by a factor of two or more, which seems significant.

*Response to comment No. 29:*
Similar to comment No. 22, we will rewrite this statement such that "consistent" is used to compare the years and aim to use other descriptions for the comparison between participants.

*Comment No. 30:*

Line 488: "Benefit of maximising income instead of power gain"
   Which results in this study demonstrate this conclusion?

*Response to comment No. 30:*
Considering your previous comments, we understand why one can criticise this statement as being too bold. However, maximising the income can be beneficial even at times with low achievable power gains if the electricity prices are high. This applies in particular for dominant wind conditions. In the revised version, we will provide more specific conclusions and add a dedicated section for future work.

**References**

[1] Göçmen, T. *et al. FarmConners Wind Farm Flow Control Benchmark: Blind Test Results, Wind Energy Science Discussions, doi: 10.5194/wes-2022-5.*

[2] *Doekemeijer, Bart M et al.. Closed-loop model-based wind farm control using FLORIS under time-varying inflow conditions, Renewable Energy, https://doi.org/10.1016/j.renene.2020.04.007*

[3] *Ott, S. and Nielsen, M. Developments of the offshore wind turbine wake model Fuga, DTU Wind Energy E-0046, DTU Wind Energy, url: https://orbit.dtu.dk/en/publications/developments-of-the-offshore-wind-turbine-wake-model-fuga*

[4] *NREL, FLORIS. Version 2.4,GitHub repository, url: https://github.com/NREL/floris*

[5] *Gebraad, P.M.O et al. Wind plant power optimization through yaw control using a parametric model for wake effects—a CFD simulation study, Wind Energy, doi:10.1002/we.1822*

---

## Author Response (AR2)

**Response to the reviewers**

Konstanze Kölle *et al.*

26 September 2022

**Reviewer 1**

*Comments to the Author:*

This paper presents an interesting study wherein different participants model the benefits of turbine or farm control in a market context and compare the impact both on energy production as well as expected revenue. I think it is a really useful exercise, to see these comparisons made in an open-access journal, and using separate research departments, models and controllers adds a degree of internal variation that is really helpful in assessing how particular the results are to any specific set of assumptions.

Reading the paper I didn't identify too much to critisize. I therefore have just a general comments.

*Comment No. 1:*
My main thought was that some of the comparisons might benefit from some extra context. In the case of energy gain, I was interested to see (and apologies if I simply missed it) some information on how each model predicts the overall starting wake losses for the two farm types. In other words the percentage difference in AEP for the farm with and without wakes.

*Response to comment No. 1:* We provided the percentage wake loss in Table 4 (p. 16 in the revised manuscript) utilising the available results from the participants for the simulated showcase. The simulations were performed with and without wind farm flow control. The latter provides the predicted power output with each wake model which was related to the theoretical power output without wakes using the turbine's power curve and the inflow wind speed. (Since the data set does not cover a whole year and thus, the wake loss was only computed for the high-prices showcases.)

In ll. 438-440 of the document with tracked changes, we referred to Table 4 when discussing the results.

*Comment No. 2:*
Then for additioinal context on the revenue side, what would be the overall

initial expected revenue of the farm. What I would be very curious to see if it would be possible would be table where each case study (P2-P5) is a row with columns like: annual revenue, delta revenue, annual profit (if it were possible to assume some cost model), expected delta profit. Was just wanted to see a rough gauge on the significance in these changes in revenue.

*Response to comment No. 2:*
We agree that the annual revenue and profit are very interesting measures. However, for the current study, the available data from P2-P5 cover only a fraction of a year due to the design of the showcases which was decided earlier. As we cannot provide annual metrics here, we have to leave those to future studies. To be more specific in the present article, we renamed it into *'FarmConners Market Showcases: Wind farm flow control considering electricity prices'*, i.e., removing the 'and revenue' from the title. Likewise, we reviewed all occurrences of the term 'revenue' to make sure it is used consistently.

*Comment No. 3:*
Finally, I had a question on comparing the EUR in 2020 versus 2030. I'm not an expert in economics, but was wondering, is there a need to control the value of the EUR between these 10 years, so that both are expressed in 2020 EUR or similar?

*Response to comment No. 3:*
The prices are actually in € 2012 value complying with the considered meteorological data from 2012. We included a clarifying comment about 2012 being the reference year for the electricity price in the revised version (ll. 59-60 in the document with tracked changes). The energy scenarios including e.g. radial offshore connections in 2030, however, differentiate 2020 from 2030.

**Reviewer 2**

**Overall comments**

The submitted article is a part of a series of papers related to the FarmConners project. Five different wind farm control methods (one individual turbine control, four wind farm flow control) are used to estimate the potential revenue increases in single 2020 and 2030 electricity price scenarios.

Overall, the motivation and contributions of the study could be more clearly explained. Results from different groups are analyzed but are not compared in detail to one another. If the focus isn't to compare the different strategies in detail, why are five distinct participants included? The stated focus of the article appears to be on evaluating methods for maximizing revenue, but the four flow control methods seem to maximize energy and the results are translated into energy after, using averaged electricity prices per wind direction sector. Also,

since the authors focus on arbitrarily defined "high price" instances, it is hard to get a sense of the actual potential to increase yearly/farm lifetime revenue. Uncertainties (of which there are many) are discussed, which is great, but the discussion is limited to only one paragraph in the conclusions.

Overall, I believe this paper could have potential to provide insight into the market potential for wind farm control but I encourage the authors to consider the comments below, in the hope that they can help improve the paper. In particular, some decisions in the methods and discussions in the results seem qualitative or ad hoc, and could use more justification.

*Response to overall comments:*
The study is the first that includes electricity prices in the evaluation of wind farm flow control. A higher share of renewable energy sources in the energy mix will provide additional incentives beyond power maximisation. Wind farms could, for example, participate in balancing markets and provide frequency support to the power system.

There are uncertainties related to the general benefit of wind farm flow control. The differences in academic implementations alone include various flow models, different control strategies and algorithms, and tuning procedures [1]. Including several participants with their different codes represents this variability. However, a comparison of farm flow models is provided, e.g., in [1].

In summary, the motivation of the current study is

- to estimate the general benefit that market-driven wind farm flow control potentially has,

- to investigate the readiness or state of the art of wind farm flow control for participation in electricity markets,

- reveal the gaps of market-driven wind farm control including farm-wide flow phenomena.

We adjusted the summary in ll. 39-47 in the document with tracked changes accordingly.

**General comments**

*Comment No. 1:*
I am not clear on why the authors have parsed the price timeseries to only consider "high prices" (arbitrarily set at the highest 25% of prices). The electricity prices don't have any impact on the power gain. But since the paper filters the wind conditions by the electricity prices for the different years, the wind conditions analyzed are different, since the electricity price timeseries are different between the two years considered. I would have thought that the "showcase" should focus on annually averaged metrics, or even better, farm lifetime averaged metrics (with uncertainty). I also think that looking at only one 2030

scenario of prices is pretty limiting, since there is substantial uncertainty around future energy prices.

*Response to comment No. 1:*

| Showcases set | Performance indicators |
| --- | --- |
| 1) High prices | income gain |
| 2) Low prices | income gain (+ load alleviation as restriction) |
| 3) TSO-driven | load alleviation, index for tracking of reference from TSO |

Table 1: Performance indicators per showcase set. TSO : transmission system operator

The showcases are built from a time series of a whole year. Originally, three showcases sets were defined as shown in Table 1. The one-year time series was split into these three showcases sets to provide incentives to not maximise power at any time. Without additional constraints, power maximisation would be the primary objective throughout the whole year. The *2) Low prices* and *3) TSO-driven* cases have performance indicators different from income gain only.

It was up to the participants to decide in which showcases they wanted to participate and how they would perform the simulations. All participants chose showcase *1) High prices*, and to maximise power in order to increase the income. There were no participants for the other two showcases.

This shows not least the current state of the research field focusing on power maximisation. In the revised version, we will elaborate on the background of the showcases sets and the choice by the participants to emphasise the conclusion that the majority of research codes for wind farm control is not ready to balance several objectives.

The changes you can see in Sections 1.1 and 1.2 were performed to better explain this. In addition, we clarified the related future work in ll. 551-554 of the document with tracked changes.

We agree that considering annual or lifetime metrics will be important in the long term to evaluate revenue-maximising strategies. However, it is not directly beneficial for the current study which – despite the discussed limitations – is the first to analyse electricity prices in the evaluation of wind farm flow control starting with two energy scenarios. Future studies should investigate the impact of uncertain electricity prices in various future scenarios.

*Comment No. 2:*
The study doesn't appear to actually do any revenue maximization, which I believe would have been much more interesting. The results demonstrated in this study take energy gain and multiply by wind direction sector averaged energy prices. But maximizing revenue (balancing short-term revenue gains from power increases and long-term (potential) revenue gained/lost due to decreased/increase fatigue loads (causing changes in O&M, different lifetime, etc.))
could potentially lead to different control strategies, which would be interesting
for the community.

*Response to comment No. 2:*
This statement is partly true. Revenue maximisation has been done for the
single wind turbine whereas it was not maximised for a wind farm, *per se*.

We agree that the outlined revenue maximisation balancing short- and long-
term gains would be very interesting and of great use for the research community.
However, such analysis requires much longer time-series simulations. This is,
as also discussed in the reply to general comment No. 1, not within the scope
here.

To meet the criticism, we renamed the article into *'FarmConners Market
Showcases: Wind farm flow control considering electricity prices'*, i.e., remov-
ing the 'and revenue' from the title.

**Point comments**

*Comment No. 1:*
Lines 2-4: "For this, offshore wind will play a major role, significantly con-
tributing to a paradigm shift in the power generation and greater volatility of
electricity prices. The operating strategy of wind farms should therefore move
from a power maximization to revenue maximisation design."

I don't quite follow the logic in these two sentences and I would suggest
rephrasing. Why would a greater volatility of electricity prices necessitate rev-
enue maximization rather than power maximization? My initial thought after
reading the first sentence was that farms should focus on system benefits (e.g.
regulation services) rather and power maximization or revenue maximization.

*Response to comment No. 1:*
In case of low electricity prices, it could be beneficial for the wind farm operator
to produce less power than available. Such operation can contribute to the
power reserve and it is possible to choose operational points where less fatigue
is accumulated reducing O&M costs on the long run. The turbines can be rather
'sacrificed' in situations with high prices when more income can be generated.

To reflect this, we rephrased the sentences into: "For this, offshore wind will
play a major role, significantly contributing to a paradigm shift in the power
generation and greater volatility of electricity prices. The operating strategy of
wind farms should therefore move from power maximisation to revenue maximi-
sation design profit maximisation which includes income from providing power
system services and the reduction of maintenance costs. (ll. 2-5 in the document
with tracked changes)

*Comment No. 2:*
Line 12: "[. . . ] and a favourable control strategy for dominant wind directions

can pay off."

This statement is a bit vague. Do the authors mean that a revenue maximizing strategy differs from a power maximizing strategy? It would also help to include quantitative statements in the abstract.

*Response to comment No. 2:*
On the long term, a revenue maximising strategy differs from a power maximising strategy. However, the statement here was that a large power gain for dominant wind directions will pay off even if the electricity prices are low. To clarify this, we changed the respective sentence to: "[...] and a favourable control strategy for dominant wind directions can pay off **even for low electricity prices**". (ll. 13-14 in the document with tracked changes)

*Comment No. 3:*
Section 1.2: I did not quite follow the strategy to determine energy price scenarios. It appears that 2020 and 2030 price scenario timeseries are generated. Then the price timeseries in parsed into low and high prices, corresponding to the lowest and highest 25% of the price data respectively? Please add a few more sentences explaining and justifying these selections. Why was no full year scenario run with all of the price data (rather than just 25% of the highest prices)?

*Response to comment No. 3:*
The division into three showcases sets with different control objectives should artificially introduce incentives for the wind farm operator to not maximise the power at any time.

We explained the idea of the original showcases sets in the revised paper with more details in the last paragraph of Section 1.1 (from l. 69 in the document with tracked changes). Please see also our response to the general comment No. 1.

Please note that Participant 1 used the whole one-year time series for the simulations of a single wind turbine.

*Comment No. 4:*
Line 83: Define power-boosting for individual turbine control.

*Response to comment No. 4:*
Power boosting for the individual turbine is based on changing the controller's set points in terms of blade pitch, generator torque and/or generator speed effectively changing the power output. This is already offered by some manufacturers. We added a reference to an OEM product to explain this better (from line 103 in the document with tracked changes). In this case, the power boosting is performed by increasing the rated torque and keeping the rated speed set-point constant. The respective description was updated in lines 105-107.

*Comment No. 5:*
Line 94: Can the authors include more details and references for the P1 aeroelastic and surrogate modeling methodology?

*Response to comment No. 5:*
We edited the explanations starting from l. 111 in the document with tracked changes. To be more clear about the aeroelastic code, we mentioned that FAST was used already in ll. 137-138.

*Comment No. 6:*
Line 98: What is meant by "desired trajectory"?

*Response to comment No. 6:*
The desired trajectory refers to the chosen set points assigned to the controller in order to achieve the different power output values. It is the trajectory followed within the $C_p - \lambda - \theta$ maps. There are multiple combinations of tip speed ratio and pitch angle that can produce the same $C_p$ output with different trade-offs in terms of loads (and power in control region 1.5). We rephrased the text starting from line 118 in the document with tracked changes to: "(...) and power-boost modes. This is achieved by changing the set-points for power coefficient ($C_p$), tip speed ratio ($\lambda$) and blade pitch angles ($\theta$). The desired set-point trajectories for the modes are identified from $C_p - \lambda - \theta$-maps, which are (...)"

*Comment No. 7:*
Line 100: "The whole process is automated so that given a desired trajectory, the relevant design variables can be estimated, i.e., the torque constant, rated values, cut in, and rated wind speed."

I did not follow this sentence. What do the authors mean by estimating the relevant design variables? Are the authors considering revenue maximizing wind turbine design in this study? My understanding was that this study focused on wind farm flow control (WFFC), and there is an existing reference case with fixed turbine properties. Perhaps these variables need to be dynamically estimated because of the modified control? It's not clear to me why you would need to estimate these design variables if you have "power coefficient (Cp), tip speed ratio, and blade pitch angle" already.

*Response to comment No. 7:*
Together with the changes applied to reply to comment No. 6, the sentence above should be clearer for the reader.

*Comment No. 8:*
Figure 2: This is a helpful figure. The authors could emphasize that only turbine pitch is being modified to achieve the controller mode.

*Response to comment No. 8:*
Thanks for the positive feedback! Besides the pitch angles, the torque controller set points have to be changed, too. For example, to achieve down-regulation in below rated conditions we need to adjust the torque constant according to the

desired power coefficient. Moreover, figure 2 explains the general methodology and could be achieved with any other trajectory achieving the same $C_p$ with other possible trade-offs in structural loading and actuator usage. However, the adjustments related to the previous comments support a better understanding of the method for the reader.

*Comment No. 9:*
Section 2: To ensure a self-contained article, the authors should briefly describe each controller mode shown in Figure 2.

*Response to comment No. 9:*
The methodology to adjust set points for down-regulation and power boosting is described from l. 125 in the document with tracked changes. The concept of changing set points has been discussed in the literature, with relevant studies already cited (Astrain Juangarcia et al., 2018; D.C. van der Hoek; and Kanev, 2017). The IBC loop is decoupled from the collective pitch controller and is explained in detail in the cited work (Pettas and Cheng 2018). The clearer explanation of the method following from adjustments related to the previous comments together with these citations describe the controllers.

*Comment No. 10:*
Line 118: How long does it take to achieve steady-state behavior in the FAST simulations? Is this hysteresis neglected in the timeseries evaluation of the controller?

*Response to comment No. 10:*
The simulations described are turbulent and therefore do not achieve steady behaviour. The duration of analysed simulations is one hour. In case the reviewer refers to the possible initial transients of the simulations, this is taken care of by using the circular wind fields created by TurbSim which allow the same wind field to be repeated, leading to the same statistical properties when a part is removed. The duration of the wind field was 3600 s and the simulation time was 3700 s with the first 100 s removed. This was clarified from l. 141 in the document with tracked changes.

*Comment No. 11:*
Figure 3:

- a. Perhaps make it clear in the caption that some figures show a subset of the 2030 timeseries while others show the full timeseries.

- b. Bottom left subfigure: Should the y-axis be "instantaneous power" instead of "rated power"?

- c. Bottom right subfigure: This plot could use more explanation. Why is there a large initial transient in each case? I also find the terminology "Accumulated DEL" confusing when it is being computed using Eq. (1).

*Response to comment No. 11:*

    a. The caption as updated according to the suggestion by the reviewer

    b. This is not the instantaneous power of the turbine but the nominal power (as percentage from the turbine rating) that shows whether we are in down-regulation or power boosting mode. E.g. in 9 m/s wind speed when the rating is 9 MW the turbine produces 90% of the power compared to the baseline. We replaced the y-label with Power Output in percentage (where 100% corresponded to a 10 MW rated power)

    c. The transient, in the beginning, comes because we start with 0 DEL. In the beginning, every small change contributes more to the DEL according to the number of events (see eq 1). As the accumulation continues the weight of each added DEL is smaller making the rate of change slower in time. Moreover, at the beginning of the calendar year (January), wind speeds are generally higher due to the winter season, the DELs are in general higher than in the following months leading to this temporary overshoot of the accumulated value. We changed the y-axis label to cumulative DEL to match equation 1 and added an explanation from l. 174 in the document with trackd changes.

*Comment No. 12:*
Line 166: Can the authors provide more details of the tuning process to give insight into challenges and research gaps? What input variables are the authors setting user-defined thresholds on? Just the PI pitch controller?

*Response to comment No. 12:*
The tuning process involves parameters such as the amount of price/wind speed bins used, low-revenue threshold to shut down the turbine, upper revenue threshold to go into maximum full boost, etc. In the optimization itself, the weights of the objective function and the penalization of values had to be tuned differently for each of the optimization objectives shown. For this initial application, this was done manually by trial and error and a methodology to cover how this can be automated is not implemented.

    The PI pitch controller is neither input nor output variable in the optimization process. The inputs are the price/speed values and the surrogates. The output is the power output level to be used per wind speed-price bin to achieve cumulative DEL and revenue objectives.

    This paper does not focus on the analysis of this method. By showing an initial and simple application of the suggested process we want to demonstrate that flexible turbine operation, considering electricity prices as input and revenue and fatigue as objectives, can lead to improvements in long-term objectives. The presented degree of details is analogous to the level of detail provided about the wind farm control strategies in Section 3. More insight on the detailed parameters, challenges and research gaps of the wind turbine controller will be

included in a dedicated publication that is currently under preparation.

*Comment No. 13:*
Line 170: This paragraph's discussion and the results in Figure 5 are very interesting! It would be great if the authors could provide more discussion about challenges and opportunities for multi-objective optimization of revenue and loads based on your results (as also related to the previous comment). It appears from the discussion that the empirical results shown here depend strongly on the tuning.

*Response to comment No. 13:* As in our previous answer, we agree with the reviewer that this discussion is interesting. Nevertheless, we believe that this discussion would have limited benefit here since the current paper has a different scope.

*Comment No. 14:*
Figure 5 is very small, consider increasing the figures and the fontsize.

*Response to comment No. 14:* The figure was enlarged according to the reviewer's recommendation.

*Comment No. 15:*
Line 175: This statement makes it more clear why the details were very limited here. No reference is provided for this method, just an indication of a future study. This is not ideal, as articles should be self-contained, or at a minimum have adequate descriptions and references to peer-reviewed publications which are accessible to readers.

*Response to comment No. 15:* We added the presentation from the WESC 2021 conference (zenodo.org/record/5017956) to the references, where this method was initially introduced. Another manuscript is currently under preparation focusing exclusively on the single turbine optimization of revenue and loads including controller tuning, surrogate modelling and optimization approaches with more realistic preview horizons. As this is a novel field of research there are no other references we are aware of. Nevertheless, we believe that the descriptions included in the manuscript are adequate for the reader to understand the general methodology. The controller design and surrogate model approach are explained and combined with references. Their combination and application to the simulation and optimization framework are explained and illustrated. The optimizers inputs for this study are based on trial and error, a method which hardly can be further elaborated. The wind farm control strategies in Section 3 are explained to a similar degree of detail.

*Comment No. 16:*
Table 2:

a. I recognize 'Gauss-legacy' is the FLORIS terminology, and useful to include, but can the authors explain in the table caption why it is called 'legacy' to inform readers who aren't as familiar with FLORIS.

b. P3: Citation missing for linear superposition. Is this linear superposition of deficits with respect to freestream or the local velocity [1]?

c. P2 and P4 are using identical model setups?

*Response to comment No. 16:*

a. Thanks for the suggestion, the following comment is now included in the table caption:

   *The term Gauss-legacy is used in FLORIS Version 2 as opposed to the Gaussian formulation implementing the near wake model by (Blondel and Cathelain,2020), as described in (Fleming et al., 2020).*

b. Linear superposition is intrinsic to Fuga [3] because it is a linearized solution to a non-linear forcing. PyWake superimposes deficits calculated for each individual turbine, respect to the local free stream, which takes into account any wake deficit from upstream turbines, if any. The reference was added to the manuscript.

c. The underlying models are the same (Table 2), but the parameters tuning is different, as can be seen in Table 3. P4 has used the default setting of FLORIS, with default parameters in the FLORIS code [4].

*Comment No. 17:*
Table 3:

a. Add a definition of each parameter introduced in this table. It would be much easier for readers to make this paper more self-contained, rather than having to refer to FLORIS documentation (which is also not archival since GitHub repositories can evolve).

b. Why are different values of n used by P2 and P4? These values of n depend on the turbine model, the wind speed and direction shear [2], and the waked conditions [3]. Since the test case is the same between the participants (with DTU-10MW turbines), I would have expected n to be the same.

*Response to comment No. 17:*

a. We included this.

b. P2 and P4 used different procedures for tuning. P4 has used the default value $n = 1.88$ in FLORIS, which was determined based on high fidelity CFD simulations by Gebraad et al. in [5]. This was clarified from l. 280 in the document with tracked changes.

*Comment No. 18:*

Section 3.3: How are the wind shear exponents found? Please include more details. To which data are the power-law curves fit? Please also explain how the wind shear exponent is used by the wake models in FLORIS.

*Response to comment No. 18:*

In P4's simulation, the wind shear exponent was derived by fitting the power-law curve with mean wind speeds at two heights: 50 m and 150 m. This also means that the wind shear exponent for each flow case is derived independently based on the provided measurement data. This has been described in section 3.3.1 by:

*Note that for simulating yawed and non-yawed wind farm production for each inflow bin, the wind shear exponent was derived with a power law based on the mean wind speed at two heights: 50 m and 150 m,*

In FLORIS, the wind shear exponent is used to model the 3D background flow field with regards to a specific hub height wind speed. Details can be found in the implementation of the *_compute_initialized_domain* function of the Flow-Field class in the Python script *flow_field.py* of the FLORIS code [4].

*Comment No. 19:*

Section 3.4: What is the power-yaw exponent for P5?

*Response to comment No. 19:*

As mentioned in section 3.4.2, the participant P5 uses the same expression for turbines under yaw as [2] for the DTU 10MW. The BEM derived expression for power coefficient under yawed conditions, i.e. $C_P(\gamma) = C_P * cos^3\gamma$, is multiplied by a scaling factor $\eta(\gamma) = \frac{1.08}{cos\gamma}$. This was specified by the addition in l. 320 in the document with tracked changes.

*Comment No. 20:*

Figure 6: I suggest having the same colorbar axis limits for all of the subfigures.

*Response to comment No. 20:* We considered using the same axes limits when preparing the original draft. However, it was very difficult to see the results for P2 with the axis limits [0; 10], so we concluded that it would be better to use different bars.

*Comment No. 21:*

Line 346: "Adjacent wind speed bins with high and low power gains in the same sector are observed exclusively in P5 results."

This also appears to happen with P3 (e.g. northerly flow), or am I misunderstanding this statement?

*Response to comment No. 21:*

The same observation can be made for P3. However, the difference observed for adjacent bins ($\sim$2% vs. $\sim$10%) is larger for P5. We updated the statement

*Comment No. 22:*
Line 348: "The discussed Figures 6 and 7 show consistent power gains per participant for both years."

What do the authors mean by "consistent"? Consistent between 2020 and 2030 or consistent between the different participants. If it is the latter, I would suggest the results have significant spread.

*Response to comment No. 22:*
The intention was to describe the consistency between the two years. The sentence was rephrased to: "The discussed Figures 6 and 7 show consistent power gains between 2020 and 2030 for each participant." (ll. 382-383 in the document with tracked changes)

*Comment No. 23:*
Line 373: "Although both P3 and P4 simulated the full TC-RWP, the different control strategies applied are seen to be the main driver for that disparate behaviour."

Can we definitively say this or can it also relate to the different wind farm model?

*Response to comment No. 23:*
We cannot exclude this and therefore rephrased the statement to: "Both P3 and P4 simulated the full TC-RWP using different control strategies applied and flow models. The difference in control strategies is considered to be the main driver for that disparate behaviour even though the different flow models will also have an impact."

*Comment No. 24:*
Line 384: "The energy gain reported by P5 in 2020 is 2%, which is slightly higher than that in 2030 with 1.7%."

Are these differences (between 1.7% and 2%) statistically significant?

*Response to comment No. 24:*
Given the other uncertainties, this is not statistically significant. We removed the values and wrote from l. 420 in the document with tracked changes: "The energy gain reported by P5 in 2030 is also about the same as that in 2030."

*Comment No. 25:*
Figure 8:

a. Is the high energy gain associated with P4 due to the discretization of the wind directions within a wind direction bin?

b. With this (and other bulk metric figures) it probably makes sense to plot

the income gain per turbine so that P2 isn't falsely interpreted as an outlier.

*Response to comment No. 25:*

a. Yes, the discretization method taken by P4 tends to yield a more optimistic energy gain, since the yaw angles are optimized for each degree inside the wind direction bins. This has been discussed in the last paragraph of section 3.3.2, as:

   *By considering 31 flow cases with different wind directions and solving the yaw optimisation problem separately for each inflow bin, P4 took an idealised or 'greedy' approach that tends to explore the full potential of wake steering, since the effectiveness of wake steering can be quite sensitive to the wind direction. However, in real-life implementation, limits on the speed and accuracy of the yawing system, uncertainty of the measured inflow wind direction, and other factors can make the reported energy gain hard to be fully realised.*

b. Following the advice to report values per wind turbine, we included some values in ll. 438-440 in the document with tracked changes.

*Comment No. 26:*
Line 395: How is income gain computed? Is the power increase for each time step multiplied by the cost of electricity within that timestep? In the introduction, the study is motivated by the time-varying nature of the energy prices, so it would seem natural to do a timeseries analysis rather than using a mean energy price per wind direction sector.

*Response to comment No. 26:*
Yes, the power increase per bin is multiplied with the electricity price. Please see also the response to general comment No. 1 and the related changes in Sections 1.1 and 1.2. The showcases sets (including the one with high prices) were defined to analyse the impact of electricity prices on the outcome using wind farm flow control. A pure time series analysis was not feasible in this context.

*Comment No. 27:*
Line 403: Sentence starting "Accordingly [...]" is a bit confusing. I didn't understand the point being made. Consider rephrasing.

*Response to comment No. 27:*
The sentence was rephrased starting in l. 443 in the document with tracked changes to: "The variability in prices among the bins is low for the investigated high-prices showcase as shown in Figure 1. Accordingly, the income gain in Figures 10 and 11 reported per participating model during the high prices is mainly driven by the estimated power gain."

*Comment No. 28:*

Line 469: "The benefit of revenue maximisation and structural load reduction as control objectives depending on the electricity prices is demonstrated at a single wind turbine"

As in previous comments, I don't think the authors have actually done revenue maximization. They have done power maximization and then translated the results into economic terms through a price for energy. I expect that a revenue maximization approach would focus on balancing short-term revenue gains from power increases and long-term (potential) revenue gained/lost due to decreased/increase fatigue loads (causing changes in O&M, different lifetime, etc.).

*Response to comment No. 28:*

As stated in the reply to the previous comments you refer to, we changed the wording for a consistent use of the term "revenue".

*Comment No. 29:*

Line 482: "The normalised gains of the four analysed WFFC implementations is consistent across the simulated cases."

Not sure I agree with this qualitative statement that the different participants have "consistent" results but since it is qualitative it is subjective. It appears that the results differ in some cases by a factor of two or more, which seems significant.

*Response to comment No. 29:*

Similar to comment No. 22, we rewrote this statement such that "consistent" is used to compare the years and used other expressions to describe the comparison between participants.

*Comment No. 30:*

Line 488: "Benefit of maximising income instead of power gain"
    Which results in this study demonstrate this conclusion?

*Response to comment No. 30:*

Considering your previous comments, we understand why one can criticise this statement as being too bold. However, maximising the income can be beneficial even at times with low achievable power gains if the electricity prices are high. This applies in particular for dominant wind conditions. In the revised version, we provided more specific conclusions and added a paragraph (starting from l. 541 in the document with tracked changes) dedicated to future work.

**References**

[1] Göçmen, T. *et al. FarmConners Wind Farm Flow Control Benchmark: Blind Test Results, Wind Energy Science Discussions, doi: 10.5194/wes-2022-5.*

*[2] Doekemeijer, Bart M et al.. Closed-loop model-based wind farm control using FLORIS under time-varying inflow conditions, Renewable Energy, https://doi.org/10.1016/j.renene.2020.04.007*

*[3] Ott, S. and Nielsen, M. Developments of the offshore wind turbine wake model Fuga, DTU Wind Energy E-0046, DTU Wind Energy, url: https://orbit.dtu.dk/en/publications/developments-of-the-offshore-wind-turbine-wake-model-fuga*

*[4] NREL, FLORIS. Version 2.4,GitHub repository, url: https://github.com/NREL/floris*

*[5] Gebraad, P.M.O et al. Wind plant power optimization through yaw control using a parametric model for wake effects—a CFD simulation study, Wind Energy, doi:10.1002/we.1822*

---

## Author Response (AR3)

Dear Sara,

Thank you for handling our manuscript.

I could not find any other instructions except uploading the files here. Please let me know if there is anything else we should adjust.

Thank you!

Best regards

Konstanze